# Weighted Distillation with Unlabeled Examples

**Fotis Iliopoulos**
Google Research
fotisi@google.com

**Vasilis Kontonis**
Google Research
kontonis@google.com

**Cenk Baykal**
Google Research
baykalc@google.com

**Gaurav Menghani**
Google Research
gmenghani@google.com

**Khoa Trinh**
Google Research
khoatrinh@google.com

**Erik Vee**
Google Research
erikvee@google.com

## Abstract

Distillation with unlabeled examples is a popular and powerful method for training deep neural networks in settings where the amount of labeled data is limited: A large "teacher" neural network is trained on the labeled data available, and then it is used to generate labels on an unlabeled dataset (typically much larger in size). These labels are then utilized to train the smaller "student" model which will actually be deployed. Naturally, the success of the approach depends on the quality of the teacher's labels, since the student could be confused if trained on inaccurate data. This paper proposes a principled approach for addressing this issue based on a "debiasing" reweighting of the student's loss function tailored to the distillation training paradigm. Our method is hyper-parameter free, data-agnostic, and simple to implement. We demonstrate significant improvements on popular academic datasets and we accompany our results with a theoretical analysis which rigorously justifies the performance of our method in certain settings.

## 1 Introduction

In many modern applications of deep neural networks, where the amount of labeled examples for training is limited, *distillation with unlabeled examples* [8, 20] has been enormously successful. In this two-stage training paradigm a larger and more sophisticated "teacher model" — typically non-deployable for the purposes of the application — is trained to learn from the limited amount of available training data in the first stage. In the second stage, the teacher model is used to generate labels on an unlabeled dataset, which is usually much larger in size than the original dataset the teacher itself was trained on. These labels are then utilized to train the "student model", namely the model which will actually be deployed. Notably, distillation with unlabeled examples is the most commonly used training-paradigm in applications where one finetunes and distills from very large-scale foundational models such as BERT [13] and GPT-3 [5] and, additionally, it can be used to significantly improve distillation on supervised examples only (see e.g. [47]).

While this has proven to be a very powerful approach in practice, its success depends on the quality of labels provided by the teacher model. Indeed, often times the teacher model generates inaccurate labels on a non-negligible portion of the unlabeled dataset, confusing the student. As deep neural networks are susceptible to overfitting to corrupted labels [50], training the student on the teacher's noisy labels can lead to degradation in its generalization performance. As an example, Figure 1 depicts an instance based on CIFAR-10 where filtering out the teacher's *noisy* labels is quite beneficial for the student's performance.

We address this shortcoming by introducing a natural "noise model" for the teacher which allows us to modify the student's loss function in a principled way in order to produce an unbiased estimate

36th Conference on Neural Information Processing Systems (NeurIPS 2022).

of the student's clean objective. From a practical standpoint, this produces a fully "plug-and-play" method which adds minimal implementation overhead, and which is composable with essentially every other known distillation technique.

The idea of compressing a teacher model into a smaller student model by matching the predictions of the teacher was initially introduced by Buciluǎ, Caruana and Niculescu-Mizil [6] and, since then, variations of this method [20, 26, 31, 35, 39, 42, 49] have been applied in a wide variety of contexts [37, 48, 51]. (Notably, some of these applications go beyond compression — see e.g. [8, 47, 48, 52] for reference.) In the simplest form of the method [26], and using classification as a canonical example, the labels produced by the teacher are one-hot vectors that represent the class which has the maximum predicted probability — this method is often referred to as "hard"-distillation. More generally, Hinton et. al. [8, 20] have shown that it is often beneficial to train the student so that it minimizes the cross-entropy (or KL-divergence) with the probability distribution produced by the teacher while also potentially using a temperature higher than 1 in the softmax of both models ("soft"-distillation). (Temperatures higher than 1 are used in order to emphasize the difference between the probabilities of the classes with lower likelihood of corresponding to the correct label according to the teacher model.)

The main contribution of this work is a principled method for improving distillation with unlabeled examples by reweighting the loss function of the student. That is, we assign importance weights to the examples labeled by the teacher so that each weight reflects (i) how likely it is that the teacher has made an inaccurate prediction regarding the label of the example and (ii) how "distorted" the unweighted loss function we use to train the student is (measured with respect to using the ground-truth label for the example instead of the teacher's label). More concretely, our reweighting strategy is based on introducing and analyzing a certain noise model designed to capture the behavior of the teacher in distillation. In this setting, we are able to come up with a closed-form solution for weights that "de-noise" the objective in the sense that (on expectation) they simulate having access to clean labels. Crucially, we empirically observe that the key characteristics of our noise model for the teacher can be effectively estimated through a small validation dataset, since in practice the teacher's noise is neither random nor adversarial, and it typically correlates well with its "confidence" — see e.g. Figure 1. In particular, we use the validation dataset to learn a map that takes as input the teacher's and student's confidence for the label of a certain example and outputs estimates for the quantities mentioned in items (i) and (ii) above. Finally, we plug in these estimates to our closed-form solution for the weights so that, overall, we obtain an automated way of computing the student's reweighted objective in practice. A detailed description of our method can be found in Section 2.3.

Our main findings and contributions can be summarized as follows:

- We propose a *principled and hyperparameter-free* reweighting method for knowledge distillation with unlabeled examples. The method is efficient, data-agnostic and simple to implement.

- We present extensive experimental results which show that our method provides significant improvements when evaluated in standard benchmarks.

- Our reweighting technique comes with provable guarantees: (i) it is information-theoretically optimal; and (ii) under standard assumptions SGD optimization of the reweighted objective learns a solution with nearly optimal generalization.

## 1.1 Related work

**Fidelity vs Generalization in knowledge distillation.** Conceptually, our work is related to the paper of Stanton et al. [43] where the main message is that "good student accuracy does not imply good distillation fidelity", i.e., that more closely matching the teacher does not necessarily lead to better student generalization. In particular, [43] demonstrates that when it comes to enlarging the distillation dataset beyond the teacher training data, there is a trade-off between optimization complexity and distillation data quality. Our work can be seen as a principled way of improving this trade-off.

**Advanced distillation techniques.** Since the original paper of Hinton et. al. [20], there have been several follow-up works [2, 7, 31, 45] which develop advanced distillation techniques which aim to enforce greater consistency between the teacher and the student (typically in the context of distillation on labeled examples). These methods enforce consistency not only between the teacher's predictions

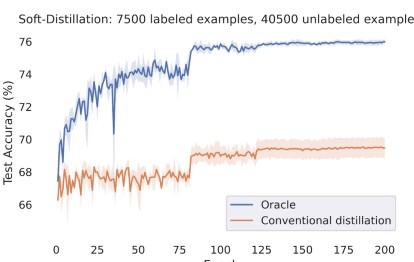
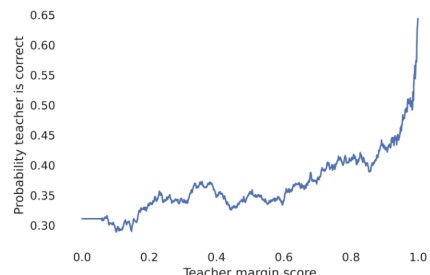

Figure 1: **Left:** Performance comparison between a student trained using conventional distillation with unlabeled examples (orange) and a student trained only on the examples which are labeled *correctly* by the teacher (blue). Here we assume access to 7500 labeled examples and 40500 unlabeled examples of CIFAR-10. The teacher is a MobileNet of depth multiplier 2, while the student is a MobileNet of depth multiplier 1. **Right:** Plot of teacher's accuracy as a function of the margin score.

and student's predictions, but also between the representations learned by the two models. However, in the context of distillation with unlabeled examples, forcing the student to match the teacher's inaccurate predictions is still harmful, and therefore weighting the corresponding loss functions via our method is still applicable and beneficial. We demonstrate this fact by considering instances of the Variational Information Distillation for Knowledge Transfer (VID) [2] framework, and showing how it is indeed beneficial to combine it with our method in Section 3.1.3. We also show that our approach provides benefits on top of any improvement obtained via temperature-scaling in Appendix B.2.

**Learning with noisy data techniques.** As we have already discussed, the main conceptual contribution of our work is viewing the teacher model in distillation with unlabeled examples as the source of stochastic noise with certain characteristics. Naturally, one may wonder what is the relationship between our method and works from the vast literature of learning with noisy data (e.g. [4, 14, 16, 23, 25, 27, 29, 33, 36, 38]). The answer is that our method *does not attempt to solve the generic "learning with noisy data"* problem, which is a highly non-trivial task in the absence of assumptions (both in theory and in practice). Instead, our contribution is to observe and exploit the fact that the noise introduced by the teacher in distillation has structure, as it correlates with several metrics of confidence such as the margin-score, the entropy of the predictions etc. We use and quantify this empirical observation to our advantage in order to formulate a principled method for developing a debiasing reweighting scheme (an approach inspired by the "learning with noisy data"-literature) which comes with theoretical guarantees. An additional difference is that works from the learning with noisy data literature typically assume that the training dataset consists of corrupted *hard* labels, and often times their proposed method is not compatible with *soft*-distillation (in the sense that they degrade its performance, making it much less effective) — see e.g. [32] for a study of this phenomenon in the case of label smoothing [28, 44].

**Uncertainty-based weighting schemes.** Related to our method are approaches for semi-supervised learning where examples are downweighted or filtered out when the teacher is "uncertain." (A plethora of ways for defining and measuring "uncertainty" have been proposed in the literature, e.g. entropy, margin-score, dropout variance etc.) These methods are independent of the student model (they only depend on the teacher model), and so they cannot hope to remove the bias from the student's loss function. In fact, these methods can be viewed as preprocessing steps that can be combined with the student-training process we propose. To demonstrate this we both combine and compare our method to the fidelity-based weighting scheme of [12] in Section 3.3.

## 1.2 Organization of the paper.

In Section 2, we present our method in detail, while in Section 3 we present our key experimental results. In Section 4, we discuss the theoretical aspects of our work. In Section 5, we summarize our results, we discuss the benefits and limitations of our method, and future work. Finally, we present extended experimental and theoretical results in the Appendix.

## 2 Weighted distillation with unlabeled examples

In this section we present our method. In Section 2.1, we review multiclass classification. In Section 2.2, we introduce the noise model for the teacher in distillation, which motivates our approach. And finally, in Section 2.3, we describe our method in detail.

### 2.1 Multiclass classification

In multiclass classification, we are given a training sample $S = \{(x_1, y_1), (x_2, y_2), \ldots (x_n, y_n)\}$ drawn from $\mathbb{P}^n$, where $\mathbb{P}$ is an unknown distribution over instances $\mathcal{X}$ and labels $\mathcal{Y} = [L] = \{1, 2, \ldots, L\}$. Our goal is to learn a predictor $f : \mathcal{X} \to \mathbb{R}^L$, namely to minimize the *risk* of $f$. The latter is defined as the expected loss of $f$:

$$R(f) = \mathbb{E}[\ell(y, f(x))] \tag{1}$$

where $(x, y)$ is drawn from $\mathbb{P}$, and $\ell : [L] \times \mathbb{R}^L \to \mathbb{R}_+$ is a loss function such that, for a label $y \in [L]$ and prediction vector $f(x) \in \mathbb{R}^L$, $\ell(y, f(x))$ is the loss incurred for predicting $f(x)$ when the true label is $y$. The most common way to approximate the risk of a predictor $f$ is via the so-called *empirical risk*:

$$R_S(f) = \frac{1}{|S|} \sum_{(x,y) \in S} \ell(y, f(x)). \tag{2}$$

That is, given a hypothesis class of predictors $\mathcal{F}$, our goal is typically to find $\min_{f \in \mathcal{F}} R_S(f)$ as a way to estimate $\min_{f \in \mathcal{F}} R(f)$.

### 2.2 Debiasing weights

As we have already discussed, the main drawback of distillation with unlabeled examples is essentially that the empirical risk minimizer corresponding to the dataset labeled by the teacher cannot be trusted, since the teacher may generate inaccurate labels. To quantify this phenomenon and guide our algorithmic design, we consider the following simple but natural noise model for the teacher.

Let $\mathbb{X}$ be an unknown distribution over instances $\mathcal{X}$. We assume the existence of a *ground-truth classifier* so that each $x \in \mathcal{X}$ is associated with a ground-truth label $f_{\text{true}}(x) \in [L] = \{1, 2, \ldots, L\}$. In other words, a clean labeled example is of the form $(x, f_{\text{true}}(x)) \sim \mathbb{P}$ (and $x \sim \mathbb{X}$). Additionally, we consider a stochastic adversary that given an instance $x \in \mathcal{X}$, outputs a "corrupted" label $y_{\text{adv}}(x)$ with probability $p(x)$, and the ground-truth label $f_{\text{true}}(x)$ with probability $1 - p(x)$. Let $\mathbb{D}$ denote the induced adversarial distribution over instances and labels.

It is not hard to see that the empirical risk with respect to a predictor $f$ and sample from $\mathbb{D}$ is *not* an unbiased estimator of the risk

$$R(f) = \mathbb{E}_{x \sim \mathbb{X}}[\ell(f_{\text{true}}(x), f(x))] \tag{3}$$

— see Proposition 2.1. On the other hand, the following *weighted* empirical risk (5) is indeed an unbiased estimator of $R(f)$. For each $x \in \mathcal{X}$ let

$$w_f(x) = \frac{1}{1 + p(x)\,(\text{distortion}_f(x) - 1)}, \quad \text{where} \quad \text{distortion}_f(x) = \frac{\ell(y_{\text{adv}}(x), f(x))}{\ell(f_{\text{true}}(x), f(x))}, \tag{4}$$

and define

$$R_S^w(f) = \frac{1}{|S|} \sum_{(x,y) \in S} w_f(x)\,\ell(y, f(x)), \tag{5}$$

where $S = \{(x_i, y_i)\}_{i=1}^n \sim \mathbb{D}^n$. Observe that the weight for each instance $x$ depends on (i) how likely it is the adversary corrupts its label; and on (ii) how this corrupted label "distorts" the loss we observe at instance $x$. In the following proposition we establish that the standard (unweighted) empirical risk with respect to distribution $\mathbb{D}$ and a predictor $f$ is a biased estimator of the risk of $f$ under the clean distribution $\mathbb{P}$, while the weighted empirical risk (5) is an unbiased one.

**Proposition 2.1** (Debiasing Weights). *Let $S = \{(x_i, y_i)\}_{i=1}^n \sim \mathbb{D}^n$ be a sample from the adversarial distribution. Defining* $\text{Bias}(f) = \mathbb{E}_{x \sim \mathbb{X}}\left[p(x) \cdot (\text{distortion}_f(x) - 1) \cdot \ell(f_{\text{true}}(x), f(x))\right]$ *we have:*

$$\textit{"reweighted"} \quad \mathbb{E}[R_S^w(f)] = R(f) \qquad vs \qquad \textit{"standard"} \quad \mathbb{E}[R_S(f)] = R(f) + \text{Bias}(f)\,.$$

Notice that, as expected, the bias of the unweighted predictor is a function of the "power" of the adversary, i.e., a function of how often they can corrupt the label of an instance, and the "distortion" this corruption causes to the loss we observe. The proof of Proposition 2.1 follows from simple, direct calculations and it can be found in Appendix D.3.

Intuitively, given a sufficiently large sample $S \sim \mathbb{D}^n$, optimizing an unbiased estimator for the risk should provide a better approximation for $\min_{f \in F} R(f)$ compared to optimizing an estimator with constant bias. We formalize this intuition in Section 4 and in Appendix D.

## 2.3   Our method

We consider the standard setting for distillation with unlabeled examples where we are given a dataset $S_\ell = \{(x_i, y_i)\}_{i=1}^m$ of $m$ labeled examples from an unknown distribution $\mathbb{P}$, and a dataset $S_u = \{x_i\}_{i=m+1}^{m+n}$ of $n$ unlabeled examples — typically, $n \geq m$. We also assume the existence of a (small) clean validation dataset $S_v = \{(x_i, y_i)\}_{i=m+n+1}^{i=m+n+q}$ of size $q$. Finally, let $\ell : \mathbb{R}^L \times \mathbb{R}^L \to \mathbb{R}_+$ be a loss function that takes as input two vectors over the set of labels $[L]$. We describe our method below and more formally in Algorithm 1 in Appendix A.

**Remark 2.1.** *The only assumption we need to make about the validation set $S_v$ is that it is not in the train set of the teacher model. That is, set $S_v$ can be used in the train set of the student model if needed — we chose to present $S_v$ as a completely independent hold out set to make our presentation as conceptually clear as possible.*

**Training the teacher.** The teacher model is trained on dataset $S_\ell$, and then it is used to generate labels for the instances in $S_u$. The labels can be one-hot vectors or probability distributions on $[L]$, depending on whether we apply "hard" or "soft" distillation, respectively.

**Training the student.** We start by pretraining the student model on dataset $S_\ell$. Then, the idea is to think of the teacher model as the source of noise in the setting of Section 2.2, compute a weight $w_f(x)$ for each example $x$ based on (4), and finally train the student on the union of labeled and teacher-labeled examples by minimizing the weighted empirical risk (examples from $S_\ell$ are assigned unit weight).

We point out two remarks. First, in order to apply (4) to compute the weight of an example $x$, we need to have estimates of $p(x)$ and $\text{distortion}_f(x)$. To obtain these estimates we use the validation dataset $S_v$ as we describe in the next paragraph. Second, observe that, according to (4), the weight of an example is a function of the predictor, namely the parameters of the model in the case of neural networks. This means that ideally we should be updating our weights assignment every time the parameters of the student model get updated during training. However, we empirically observe that even estimating the weight assignment *only once* during the whole training process (right after training the student model on $S_\ell$) suffices for our purposes, and so our method adds minimal overhead to the standard training process. More generally, the fact that our process of computing the weights is simple and inexpensive allows us to recompute them during training (say every few epochs) to improve our approximation of the theoretically optimal weights. We explore the effect of updating the weights during training in Section 3.2.

**Estimating the weights.** We estimate the weights using the Nearest Neighbors method on $S_v$ to learn a map that takes as input the teacher's and student's "confidence" for the label of a certain example $x$, and outputs estimates for $p(x)$ and $\text{distortion}_f(x)$ so we can apply (4). In our experiments we measure the confidence of a model either via the *margin-score*, i.e., the difference between the largest two predicted classes for the label of a given example (see e.g. the so-called "margin" uncertainty sampling variant [40]), or via the *entropy* of its prediction. However, one could use *any* metric (not necessarily confidence) that correlates well with the accuracy of the corresponding models.

More concretely, we reduce the task of estimating the weights to a two-dimensional (i.e., two inputs and two outputs) regression task over the validation set which we solve using the Nearest Neighbors method. In particular, our Nearest Neighbor data structure is constructed as follows. Each example $x$ of the validation set is assigned the two following pairs of points: (i) (teacher confidence at $x$, student confidence at $x$) — this is the covariate of the regression task; (ii) (1, distortion at $(x)$) if the teacher correctly predicts the label of $x$, or (0, distortion at $x$), if the teacher does not correctly predict the label of $x$ — this is the response of the regression task. The query corresponding to an unlabeled example $x'$ is of the form (teacher confidence at $x'$, student confidence at $x'$). The Nearest Neighbors

data structure returns the average response over the $k$ closest in euclidean distance pairs (teacher confidence at $x$, student confidence at $x$) in the validation set. The value of $k$ is specified in the next paragraph. The pseudocode for our method can be found in Algorithm 2 in Appendix A.

We point out two remarks. First, the number $k$ of neighbors we use for our weights-estimation is always $k = \frac{\sqrt{|S_v|}}{2}$. This is because choosing $k = \Theta(q^{2/(2+\dim)})$, where $q$ is the size of the validation dataset and $\dim$ is the dimension of the underlying metric space ($\dim = 2$ in our case), is asymptotically optimal, and $1/2$ is a popular choice for the hidden constant used in practice, see e.g. [9, 18]. Second, notice that (4) implies that the weight of an example could be larger than $1$ if (and only if) the corresponding distortion value (4) at that example is less than $1$. This could happen for example if both the student and teacher have the same (or very similar) inaccurate prediction for a certain example. In such a case, the value of the weight in (4) informs us that the loss at this example should be larger than the (low) value the unweighted loss function suggests. However, since we do not have the ground-truth label for a point during training — but only the inaccurate prediction of the teacher — having a weight larger than $1$ in this case would most likely guide our student model to fit an inaccurate label. To avoid this phenomenon, we always project our weights onto the $[0, 1]$ interval (Line 16 of Algorithm 2). In Appendix D.2 we discuss an additional reason why it is beneficial to project the weights of examples of low distortion onto $[0, 1]$ based on a MSE analysis.

## 3 Experimental results

In this section we present our experimental results. In Section 3.1 we consider an experimental setup according to which the weights are estimated only once during the whole training process. In Section 3.2 we study the effect of updating the weights during training. Finally, in Section 3.3 we demonstrate that our method can be combined with uncertainty-based weighting techniques by both combining and comparing our method to the fidelity-based weighting scheme of [12].

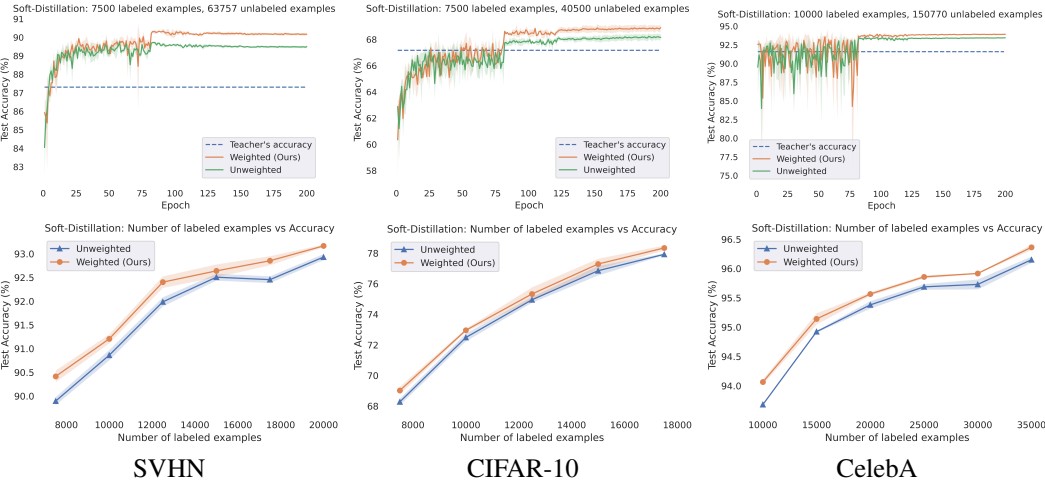

Figure 2: The student's test accuracy over the training trajectory (first row) and student's best test-accuracy achieved over all (second row) when applying **one-shot estimation of the weights**. The teacher model is a MobileNet with depth multiplier 2. In the cases of SVHN and CIFAR-10 the student model is a MobileNet of depth multiplier 1, and in the case of CelebA is a ResNet-11. Our approach leads to consistently better models in terms of test-accuracy and convergence speed. Shaded regions correspond to values within one standard deviation of the mean.

### 3.1 Improvements via one-shot estimation of the weights

Here we show how applying our reweighting scheme provides consistent improvements on several popular benchmarks even when the weights are estimated only once during the whole training process. We compare our method against conventional distillation with unlabeled examples, but we also show that our method can provide benefits when combined with more advanced distillation techniques such as the framework of [2] (see Figure 9).

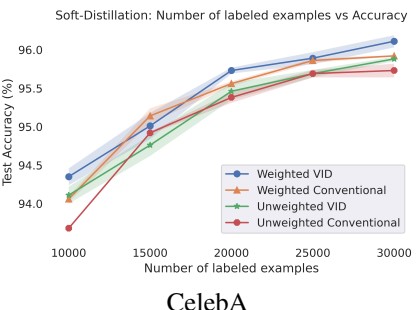
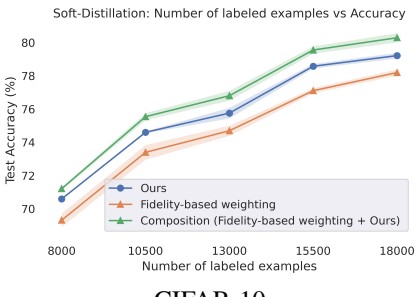

CelebA                    CIFAR-10

Figure 3: **Left:** We compare conventional distillation (unweighted conventional), standard Variational Information Distillation (VID) [2] (unweighted VID), and reweighting the conventional loss function (weighted conventional) and the VID loss function with our method (weighted VID) on CelebA. (These results correspond to **one-shot estimation of the weights**.) We see that our method can be combined effectively with more advanced distillation techniques such as VID. **Right:** We combine and compare our method with the fidelity-based reweighting scheme of [12] on CIFAR-10. We see that our method can be combined effectively with weighting schemes based only on the teacher's uncertainty. (These results correspond to **updating our weights at the end of every epoch**.)

More concretely, we evaluate our method on benchmark vision datasets. We compare our method to conventional distillation with unlabeled examples both in terms of the best test accuracy achieved by the student, and in terms of convergence speed (see Figure 2). We also evaluate the comparative advantage of our method as a function of the number of labeled examples available (size of dataset $S_\ell$). We always choose the temperature in the softmax of the models to be 1 for simplicity and consistency, and our metric for confidence is always the margin-score. Implementation details for our experiments and additional results can be found in Appendices B and C. We implemented all algorithms in Python making use of the TensorFlow deep learning library [1]. We use 64 Cloud TPU v4s each with two cores.

### 3.1.1 Experimental setup

Our experiments are of the following form. The academic dataset we use each time is first split into two parts $A$ and $B$. Part $A$, which is typically smaller, is used as the labeled dataset $S_\ell$ where the teacher model is trained on (recall the setting we described in Section 2.3). Part $B$ is randomly split again into two parts which represent the unlabeled dataset $S_u$ and validation dataset $S_v$, respectively. Then, (i) the teacher and student models are trained once on the labeled dataset $S_\ell$; (ii) the teacher model is used to generate soft-labels for the unlabeled dataset $S_u$; (iii) we train the student model on the union of $S_\ell$ and $S_u$ using our method and conventional distillation with unlabeled examples. We repeat step (iii) a number of times: in each trial we partition part $B$ randomly and independently, and then the student model is trained using the (student-)weights reached after completing the training on dataset $S_\ell$ in step (i) as initialization, both for our method and the competing approaches.

### 3.1.2 CIFAR-{10, 100} and SVHN experiments

SVHN [34] is an image classification dataset where the task is to classify street view numbers (10 classes). The train set of SVHN contains 73257 labeled images and its test set contains 26032 images. We use a MobileNet [21] with depth multiplier 2 as the teacher, and a MobileNet with depth multiplier 1 as the student[1]. The tables in Figure 4 contain the results of our experiments (averages over 3 trials). In each experiment we use the first $N \in \{7500, 10000, 12500, 15000, 17500, 20000\}$ examples as the labeled dataset $S_\ell$, and then the rest $73257 - N$ images are randomly split to a labeled validation dataset $S_v$ of size 2000, and an unlabeled dataset $S_u$ of size $71257 - N$.

CIFAR-10 and CIFAR-100 [24] are image classification datasets with 10 and 100 classes respectively. Each of them consists of 60000 labeled images, which we split to a training set of 50000 images, and a test set of 10000 images. For CIFAR-10, we use a Mobilenet with depth multiplier 2 as the teacher,

---

[1]Note that we see the student outperform the teacher here and in other experiments, as can often happen with distillation with unlabeled examples, particularly when the teacher is trained on limited data.

and a Mobilenet with depth multiplier 1 as the student. For CIFAR-100, we use a ResNet-110 as a teacher, and a ResNet-56 as the student. We use a validation set of 2000 examples. The results of our experiments (averages over 3 trials) can be found in the tables of Figures 5, 6.

| Labeled Examples | 7500 | 10000 | 12500 | 15000 | 17500 |
|---|---|---|---|---|---|
| Teacher | 87.31% | 88.5% | 88.45% | 91.38% | 91.32% |
| Weighted (Ours) | **90.41 ± 0.12%** | **91.21 ± 0.09%** | **92.4 ± 0.12%** | **92.64 ± 0.14%** | **92.85 ± 0.1%** |
| Unweighted | 89.89 ± 0.09% | 90.86 ± 0.11% | 91.99 ± 0.1% | 92.51 ± 0.07% | 92.45 ± 0.07% |

Figure 4: Experiments on SVHN. See Section 3.1.2 for details.

| Labeled Examples | 7500 | 10000 | 12500 | 15000 | 17500 |
|---|---|---|---|---|---|
| Teacher | 67.17% | 71.39% | 74.69% | 77% | 78.46% |
| Weighted (Ours) | **69.01 ± 0.25%** | **72.96 ± 0.1%** | **75.33 ± 0.43%** | **77.69 ± 0.35%** | **78.34 ± 0.16%** |
| Unweighted | 68.27 ± 0.22% | 72.48 ± 0.26% | 74.95 ± 0.2% | 76.85 ± 0.32% | 77.92 ± 0.06% |

Figure 5: Experiments on CIFAR-10. See Section 3.1.2 for details.

| Labeled Examples | 7500 | 10000 | 12500 | 15000 | 17500 |
|---|---|---|---|---|---|
| Teacher | 44.1% | 51.31% | 55.54% | 59.05% | 62.17% |
| Weighted (Ours) | **46.81 ± 0.34%** | **53.31 ± 0.66%** | **57.5 ± 0.3%** | **60.94 ± 0.32%** | **62.86 ± 0.41%** |
| Unweighted | 46.29 ± 0.04% | 52.83 ± 0.53% | 56.89 ± 0.65% | 60.73 ± 0.03% | 62.79 ± 0.09% |

Figure 6: Experiments on CIFAR-100. See Section 3.1.2 for details.

### 3.1.3 CelebA experiments: considering more advanced distillation techniques

As we have already discussed in Section 1.1, our method can be combined with more advanced distillation techniques which aim to enforce greater consistency between the teacher and the student. We demonstrate this fact by implementing the method of Variational Information Distillation for Knowledge Transfer (VID) [2] and showing how it is indeed beneficial to combine it with our method. We chose the gender binary classification task of CelebA [15] as our benchmark (see details in the next paragraph), because it is known (see e.g. [31]) that the more advanced distillation techniques tend to be more effective when applied to classification tasks with few classes. In the table below, "Unweighted VID" corresponds to the implementation of loss described in equations (2), (4) and (6) of [2], and "Weighted VID" corresponds to the reweighting of the latter loss using our method.

CelebA [15] is a large-scale face attributes dataset with more than 200000 celebrity images, each with forty attribute annotations. Here we consider the binary male/female classification task. The train set of CelebA contains 162770 images and its test set contains 19962 images. We use a MobileNet with depth multiplier 2 as the teacher, and a ResNet-11 [19] as the student. The tables in Figure 7 contain the results of our experiments (averages over 3 trials). In each experiment we use the first $N \in \{10000, 15000, 20000, 25000, 30000, 35000\}$ examples as the labeled dataset $S_\ell$, and then the rest $162770 - N$ images are randomly split to a labeled validation dataset $S_v$ of size 2000, and an unlabeled dataset $S_u$ of size $160770 - N$.

| Labeled Examples | 10000 | 15000 | 20000 | 25000 | 30000 |
|---|---|---|---|---|---|
| Teacher | 91.59% | 93.76% | 94.41% | 94.86% | 94.92% |
| Weighted VID | **94.35 ± 0.11%** | 95.01 ± 0.17% | **95.73 ± 0.04%** | **95.89 ± 0.08%** | **96.11 ± 0.08%** |
| Weighted Conventional | 94.06 ± 0.04% | **95.14 ± 0.1%** | 95.56 ± 0.03% | 95.86 ± 0.03% | 95.92 ± 0.01% |
| Unweighted VID [2] | 94.11 ± 0.11% | 94.76 ± 0.14% | 95.46 ± 0.11% | 95.69 ± 0.05% | 95.88 ± 0.03% |
| Unweighted Conventional | 93.68 ± 0.01% | 94.92 ± 0.02% | 95.38 ± 0.07% | 95.69 ± 0.05% | 95.73 ± 0.09% |

Figure 7: Experiments on CelebA. See Section 3.1.3 for details and also Figure 3.

### 3.1.4 ImageNet experiments

ImageNet [41] is a large-scale image classification dataset with 1000 classes consisting of approximately $I \approx 1.3$M images. For our experiments, we use a ResNet-50 as the teacher, and a ResNet-18 as the student. In each experiment we use the first $N \in \{64058, 128116\}$ labeled examples (5% and 10% of $I$, respectively) as the labeled dataset $S_\ell$, and the rest $I - N$ examples are randomly split to a labeled validation dataset $S_v$ of size 10000, and an unlabeled dataset $S_u$ of size $I - N - 10000$. The results of our experiments (averages over 10 trials) can be found in Figure 8.

| Labeled Examples | 5% of ImageNet | 10% of ImageNet |
|---|---|---|
| Teacher (soft) | 36.74% | 51.88% |
| Weighted (Ours) | **38.60 ± 0.07%** | **53.59 ± 0.09%** |
| Unweighted | 38.44 ± 0.06% | 53.43 ± 0.08% |

| Labeled Examples | 5% of ImageNet | 10% of ImageNet |
|---|---|---|
| Teacher (hard) | 36.74% | 51.88% |
| Weighted (Ours) | **38.56 ± 0.07%** | **53.34 ± 0.11%** |
| Unweighted | 38.42 ± 0.06% | 53.18 ± 0.07% |

Figure 8: Experiments with soft-distillation (left) and hard-distillation (right) on ImageNet.

## 3.2 Updating the weights during training

In this section we consider the effect of updating our estimation of the optimal weights during training. For each dataset we consider, the experimental setup is identical to the corresponding setting in Section 3.1, except for that we always use a validation set of size 500 and the entropy of a model's prediction as the metric for its confidence. We note that the time required for computing the weight for each example is insignificant compared to total training time (less than 1% of the total training time in all of our experiments), which allows us to conduct experiments in which we update our estimation at the end of every epoch. We see that doing this typically significantly improves the resulting student's performance (however, in CIFAR-100 we do not observe substantial benefits).

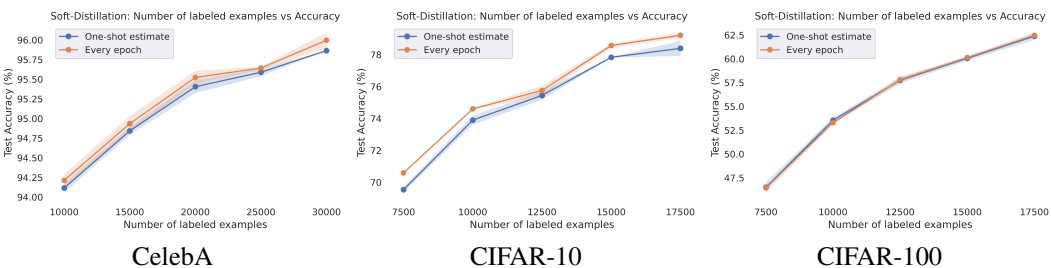

CelebA        CIFAR-10        CIFAR-100

Figure 9: The effect of updating the weights during training. We compare our method when (i) weights are estimated only once and; (ii) when we update our estimation at the end of every epoch.

## 3.3 Combining and comparing with uncertainty-based weighting schemes

As we have already discussed in Section 1.1, our method can be combined with uncertainty-based weighting schemes as these are independent of the student model and, therefore, they can be seen as a preprocessing step (modifying the loss function) before applying our method. We demonstrate this by combining and comparing our method with the filelity-based weighting scheme of [12] on CIFAR-10 and CIFAR-100. Details about our experimental setup can be found in Appendix C.3.

| Labeled Examples | 7500 | 10000 | 12500 | 15000 | 17500 |
|---|---|---|---|---|---|
| Teacher (soft) | 67.55% | 72.85% | 74.85% | 77.63% | 78.40% |
| Our Method | 70.59 ± 0.05% | 74.59 ± 0.07% | 75.75 ± 0.32% | 78.56 ± 0.14% | 79.21 ± 0.17% |
| Fidelity-based weighting [12] | 69.31 ± 0.41% | 73.39 ± 0.44% | 74.69 ± 0.31% | 77.10 ± 0.16% | 78.19 ± 0.19% |
| Composition | **71.20 ± 0.097%** | **75.54 ± 0.21%** | **76.80 ± 0.29%** | **79.55 ± 0.22%** | **80.28 ± 0.28%** |

| Labeled Examples | 7500 | 10000 | 12500 | 15000 | 17500 |
|---|---|---|---|---|---|
| Teacher (soft) | 44.70% | 51.45% | 56.00% | 58.70% | 61.82% |
| Our Method | 46.18 ± 0.25% | 53.15 ± 0.16% | **57.51 ± 0.4%** | **59.65 ± 0.4%** | 62.02 ± 0.25% |
| Fidelity-based weighting [12] | 46.32 ± 0.23% | 52.92 ± 0.20% | 57.40 ± 0.47% | 59.48 ± 0.17% | 61.39 ± 0.09% |
| Composition | **46.62 ± 0.44%** | **53.39 ± 0.16%** | 57.11 ± 0.41% | 59.63 ± 0.15% | **62.30 ± 0.26%** |

Figure 10: Combining and comparing our method with the weighting scheme of [12] on CIFAR-10 (top table) and CIFAR-100 (bottom table). See also Figure 3.

## 4 Theoretical motivation

In this section we show that our debiasing reweighting method described in Section 2.2 comes with provable **statistical** and **optimization** guarantees. We first show that the method is statistically consistent in the sense that, with a sufficiently large dataset, the reweighted risk converges to the true or "clean" risk. We show (see Appendix D) the following convergence guarantee.

**Theorem 4.1** (Uniform Convergence of Reweighted Risk). *Assume that $S$ is a dataset of i.i.d. "noisy" samples from $\mathbb{D}$. Under standard capacity assumptions for the class of models $\mathcal{F}$ and regularity assumptions for the loss $\ell(\cdot)$, for every $f \in \mathcal{F}$ it holds that*

$$\lim_{|S| \to \infty} R_S^w(f) = R(f) \quad and \quad \lim_{|S| \to \infty} R_S(f) = R(f) + \mathrm{Bias}(f)$$

To prove our optimization guarantees we analyze the reweighted objective in the fundamental case where the model $f(x; \Theta)$ is linear, i.e., $f(x; \Theta) = \Theta x \in \mathbb{R}^L$, and the loss $\ell(y, z)$ is convex in $z$ for every $y$. In this case, the composition of the loss and the model $f(x; \Theta)$ is convex as a function of the parameter $\Theta \in \mathbb{R}^{L \times d}$. Recall that we denote by $f_{\mathrm{true}}(x) : \mathbb{R}^d \mapsto \mathbb{R}^L$ the ground truth classifier and by $\mathbb{P}$ the "clean" distribution, i.e., a sample from $\mathbb{P}$ has the form $(x, f_{\mathrm{true}}(x))$ where $x$ is drawn from a distribution $\mathbb{X}$ supported on (a subset of) $\mathbb{R}^d$. Finally, we denote by $\mathbb{D}$ the "noisy" labeled distribution on $\mathbb{R}^d \times \mathbb{R}^L$ and assume that the $x$-marginal of $\mathbb{D}$ is also $\mathbb{X}$.

We next give a general definition of debiasing weight functions, i.e., weighting mechanisms that make the corresponding objective function an unbiased estimator of the clean objective $R(\Theta)$ for every parameter vector $\Theta \in \mathbb{R}^d$. Recall that the weight function defined in Section 2.2 is debiasing.

**Definition 4.2** (Debiasing Weights). *We say that a weight function $w(x, y_{\mathrm{adv}}; \Theta)$ is a debiasing weight function if it holds that*

$$R^w(\Theta) \triangleq \mathbb{E}_{(x, y_{\mathrm{adv}}) \sim \mathbb{D}}[w(x, y_{\mathrm{adv}}; \Theta) \ell(y_{\mathrm{adv}}, f(x; \Theta))] = R(\Theta) \,.$$

Since the loss is convex in $\Theta$, one could try to optimize the naive objective that does not reweight and simply minimizes $\ell(\cdot)$ over the noisy examples, $R^{\mathrm{naive}}(\Theta) \triangleq \mathbb{E}_{(x, y_{\mathrm{adv}}) \sim \mathbb{D}}[\ell(y_{\mathrm{adv}}, \Theta x)]$. We show (unsurprisingly) that doing this is a bad idea: there are instances where optimizing the naive objective produces classifiers with bad generalization error over clean examples. For the formal statement and proof we refer the reader to Appendix E.

*SGD on the naive objective $R^{\mathrm{naive}}(\cdot)$ learns parameters with arbitrarily bad generalization error over the "clean" data.*

Our main theoretical insight is that optimizing linear models with the reweighted loss leads to parameters with almost optimal generalization.

*Given a debiasing weight function $w(\cdot)$, SGD on the reweighted objective $R^w(\cdot)$ learns a parameter with almost optimal generalization error over the "clean" data.*

The main issue with optimizing the reweighted objective is that, in general, we have no guarantees that the weight function preserves its convexity (recall that it depends on the parameter $\Theta$). However, we know that its population version corresponds to the clean objective $R(\cdot)$ which is a convex objective. We show that we can use the convexity of the underlying clean objective to show results for stochastic gradient descent, by proving the following key structural property.

**Proposition 4.3** (Stationary Points of the Reweighted Objective Suffice (Informal)). *Let $S$ be a dataset of $n = \mathrm{poly}(dL/\epsilon)$ i.i.d. samples from the noisy distribution $\mathbb{D}$. Let $\widehat{\Theta}$ be any stationary point of the weighted objective $R_S^w(\Theta)$ constrained on the unit ball (with respect to the Frobenious norm $\| \cdot \|_F$). Then, with probability at least 99%, it holds that*

$$R(\widehat{\Theta}) \leq \min_{\|\Theta\|_F \leq 1} R(\Theta) + \epsilon \,.$$

## 5 Conclusion

We propose a principled reweighting scheme for distillation with unlabeled examples. Our method is hyper-parameter free, adds minimal implementation overhead, and comes with theoretical guarantees. We evaluated our method on standard benchmarks and we showed that it consistently provides significant improvements. We note that investigating improved data-driven ways of estimating the weights (4) could be of interest, since potential inaccurate estimation of the weights is the main limitation of our work. We leave this question open for future work.

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
