# A  Formal description of our method

In this section we present pseudocode for our method.

---

**Algorithm 1** Weighted distillation with unlabeled examples

---

**Require:** model Teacher, model Student, labeled dataset $S_\ell = \{(x_i, y_i)\}_{i=1}^m$, unlabeled dataset $S_u = \{x_i\}_{i=m+1}^{m+n}$, validation dataset $S_v = \{(x_i, y_i)\}_{i=m+n+1}^{m+n+q}$, number of weights-estimating iterations $r$
1: Train Teacher and Student on $S_\ell$
2: Use Teacher to generate labels for $S_u$ to obtain set $S_u^\ell = \{(x, \text{Teacher}(x)) \mid x \in S_u\}$
3: **for** $i = m + 1$ to $n + m$ **do**
4:     $y_i \leftarrow \text{Teacher}(x_i)$
5: $S \leftarrow \{(x_i, y_i)\}_{i=1}^{m+n}$
6: **for** $i = 1$ to $m$ **do**
7:     $w(x_i) \leftarrow 1$
8: **for** $i = 1$ to $r$ **do**
9:     $\{w(x_{m+1}), \ldots, w(x_{m+n})\} \leftarrow \text{ESTIMATEWEIGHTS}(\text{Teacher, Student}, S_v, S_u^\ell)$
10:     Train Student on $S$ using the weighted empirical risk:

$$\frac{1}{m+n} \sum_{i=1}^{m+n} w(x_i) \ell(y_i, \text{Student}(x_i))$$

---

---

**Algorithm 2** Procedure for estimating the weights

---

1: **procedure** ESTIMATEWEIGHTS( Teacher, Student, $V, D$ )
2:                                  $\triangleright V$ is the validation dataset and $D$ is the teacher-labeled dataset
3:     $U \leftarrow \emptyset, k \leftarrow \lceil \frac{1}{2} \sqrt{|V|} \rceil$
4:     **for** every $(x, y) \in V$ **do**
5:         $X \leftarrow (\text{Confidence}(\text{Teacher}(x)), \text{Confidence}(\text{Student}(x)))$
6:         **if** $\arg\max(\text{Teacher}(x)) = \arg\max(y)$ **then**:
7:             $(p, \text{distortion}) \leftarrow (0, 1)$
8:         **else**:
9:             $(p, \text{distortion}) \leftarrow \left(1, \frac{\ell(\text{Teacher}(x), \text{Student}(x))}{\ell(y, \text{Student}(x))}\right)$
10:         $Y \leftarrow (p, \text{distortion})$
11:         $U \leftarrow U \cup \{(X, Y)\}$
12:     Weights $= \varnothing$                           $\triangleright$ Initialize and empty list for the weights
13:     **for** every $(x, y) \in D$ **do**
14:         Query $\leftarrow (\text{Confidence}(\text{Teacher}(x)), \text{Confidence}(\text{Student}(x)))$
15:         $(\hat{p}, \hat{d}) \leftarrow k\text{-NN}(U, \text{Query})$     $\triangleright$ Predict $p(x)$ and $\text{distortion}_f(x)$ from the $k$ nearest neighbors of Query in $U$
16:         $w(x) \leftarrow \min\left\{1, \frac{1}{1+\hat{p}(\hat{d}-1)}\right\}$
17:         Append $w(x)$ to Weights
18:     **return** Weights

---

# B  Extended experiments

## B.1  The student's test-accuracy-trajectory

In this section we provide extended experimental results that show the student's test accuracy over the training trajectory corresponding to experiments we mentioned in Section 3.1. Notice that in the vast majority of cases our method significantly outperforms the conventional approach almost throughout the training process.

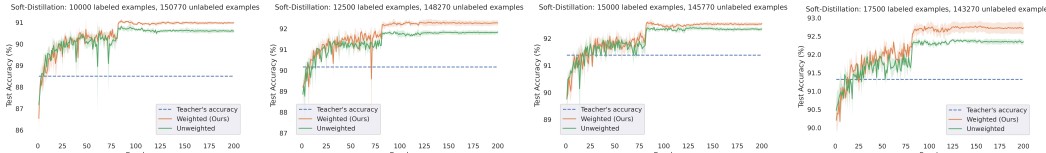

Figure 11: **SVHN** experiments. The student's test accuracy over the training trajectory using hard-distillation corresponding to the experiments of Figure 4. See Section 3.1.2 for more details.

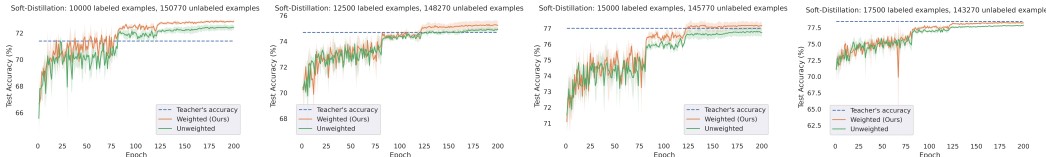

Figure 12: **CIFAR-10** experiments. The student's test accuracy over the training trajectory corresponding to the experiments of Figure 5. See Section 3.1.2 for more details.

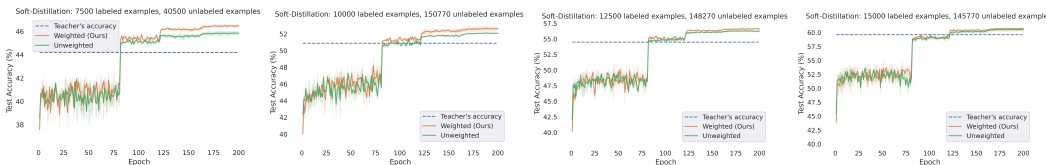

Figure 13: **CIFAR-100** experiments. The student's test accuracy over the training trajectory corresponding to the experiments of Figure 6. See Section 3.1.2 for more details.

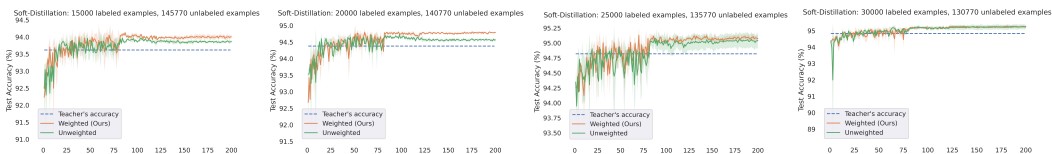

Figure 14: **CelebA** experiments. The student's test accuracy over the training trajectory corresponding to the experiments of Figure 7. See Section 3.1.3 for more details.

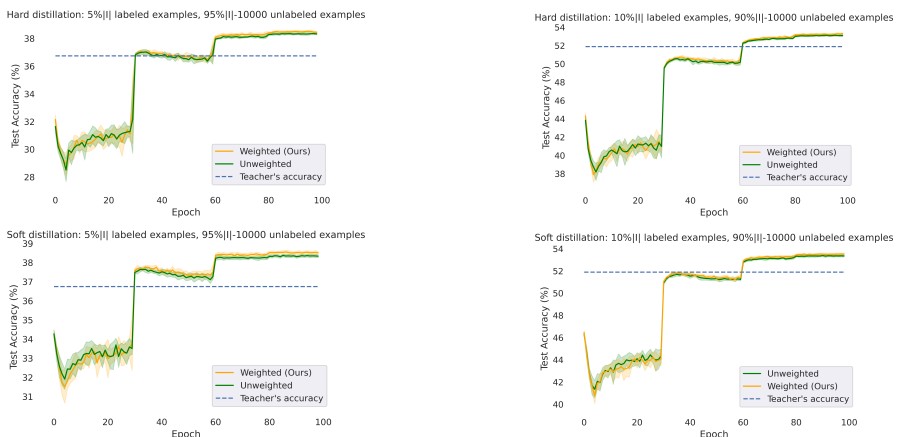

Figure 15: **ImageNet** experiments. The student's test accuracy over the training trajectory using hard-distillation (first row) and soft-distillation (second row) corresponding to the experiments of Figure 8. See Section 3.1.4 for more details.

## B.2 Considering the effect of temperature

Temperature-scaling, a technique introduced in the original paper of Hinton et. al. [20], is one the most common ways for improving student's performance in distillation. Indeed, it is known (see e.g. [43]) that choosing the right value for the temperature can be quite beneficial, to the point it can outperform other more advanced techniques for improving distillation. Here we demonstrate that our approach provides benefits on top of any improvement on can get via temperature-scaling by conducting an ablation study on the effect of temperature on CIFAR-100. In our experiment, the teacher model is a Resnet-110 achieving accuracy $56.0\%$, the student model is a Resnet-56, the number of labeled examples is $12500$, the validation set consists of $500$ examples, and we use the entropy of a prediction as a metric of confidence. We apply our method using one-shot estimation of the weights. We compare training the student model using conventional distillation to using our method for different values of temperature. The results can be found in the table below. We see that in almost all cases the student-model trained using our method outperforms the student-model trained using conventional distillation and, in particular, the best student overall is the result of choosing $2.0$ for the value of temperature and applying our method.

| Temperature | Unweighted | Weighted (ours) |
|---|---|---|
| 0.01 | $52.84 \pm 0.08\%$ | $53.73 \pm 0.11\%$ |
| 0.10 | $54.63 \pm 0.09\%$ | $54.84 \pm 0.12\%$ |
| 0.50 | $56.45 \pm 0.12\%$ | $57.01 \pm 0.1\%$ |
| 0.80 | $56.67 \pm 0.12\%$ | $57.60 \pm 0.15\%$ |
| 1.00 | $57.17 \pm 0.15\%$ | $57.56 \pm 0.09\%$ |
| 2.00 | $57.54 \pm 0.11\%$ | $\mathbf{57.8 \pm 0.21}\%$ |
| 3.00 | $57.20 \pm 0.18\%$ | $57.09 \pm 0.25\%$ |
| 5.00 | $56.92 \pm 0.11\%$ | $57.01 \pm 0.2\%$ |

Figure 16: Ablation study on the effect of temperature on CIFAR-100. See Appendix B.2 for details

# C Implementation details

In this section we describe the implementation details of our experiments. Recall the description of our method in Section 2.3.

## C.1 Experiments on CelebA, CIFAR-10, CIFAR-100, SVHN

All of our experiments are performed according to the following recipe. In all cases, the loss function $\ell : \mathbb{R}^L \times \mathbb{R}^L \to \mathbb{R}_+$ we use is the cross-entropy loss. We train the teacher model for 200 epochs on dataset $S_\ell$. We pretrain the student model for 200 epochs on dataset $S_\ell$ and save its parameters. Then, using the latter saved parameters for initialization each time, we train the student model for 200 epochs optimizing either the weighted or conventional (unweighted) empirical risk, and report its average performance over three trials.

We use the Adam optimizer. The initial learning rate is $\mathrm{lr} = 0.001$. We proceed according to the following learning rate schedule (see e.g., [19]):

$$\mathrm{lr} \leftarrow \begin{cases} \mathrm{lr} \cdot 0.5 \cdot 10^{-3}, & \text{if } \#\text{epochs} > 180 \\ \mathrm{lr} \cdot 10^{-3}, & \text{if } \#\text{epochs} > 160 \\ \mathrm{lr} \cdot 10^{-2}, & \text{if } \#\text{epochs} > 120 \\ \mathrm{lr} \cdot 10^{-1}, & \text{if } \#\text{epochs} > 80 \end{cases}$$

Finally, we use data-augmentation. In particular, we use random horizontal flipping and random width and height translations with width and height factor, respectively, equal to $0.1$.

## C.2 Experiments on ImageNet

For the ImageNet experiments we follow a similar although not identical recipe to the one described in Appendix C.1. In each training stage above, we train the model (teacher or student) for 100 epochs instead of 200. We also use SGD with momentum $0.9$ instead of Adam as the optimizer. For data-augmentation we use only random horizontal flipping. Finally, the learning rate schedule is as

follows. For the first 5 epochs the learning rate lr is increased from 0.0 to 0.1 linearly. After that, the learning rate changes as follows:

$$\text{lr} = \begin{cases} 0.01, & \text{if } \#\text{epochs} > 30 \\ 0.001, & \text{if } \#\text{epochs} > 60 \\ 0.0001, & \text{if } \#\text{epochs} > 80 \end{cases}$$

### C.3 Details on the experimental setup of Section 3.3

In the CIFAR-10 experiments of Section C.3 the teacher model is a MobileNet with depth multiplier 2, and the student model is a MobileNet with depth multiplier 1. In the CIFAR-100 experiments of the same section, the teacher model is a ResNet-110, and the student model is a ResNet-56. We use a validation set consisting of 500 examples (randomly chosen as always). The student of each method has access to the same number of labeled examples, i.e., the validation set is used for training the student model as we describe in Remark 2.1. We compare the following three methods:

- **Fidelity weighting scheme [12].** For every example $x$ we use the entropy of the teacher's prediction as an uncertainty/confidence measure, which we denote by $\text{entropy}(x)$. We then compute the exponential weights described in [12] as $w(x) = \exp(-\text{entropy}(x)/\overline{\text{entropy}})$, where $\overline{\text{entropy}}$ is the average entropy of the teacher's predictions over all training examples.

- **Our method.** We use the entropy as the metric for confidence. In the case of CIFAR-10 we re-estimate the weights at the end of every epoch. In the case of CIFAR-100 the weights are estimated only once in the beginning of the process.

- **Composition.** We reweight each example in the loss function by multiplying the weights resulting from the two methods above.

## D Extended theoretical motivation: statistical aspects

In this section we study the statistical aspects of our approach. In Section D.1 we revisit and formally state Theorem 4.1 — see Corollary D.2 and Remark D.1. In Section D.2 we perform a Mean-Squared-Error analysis that provides additional justification of our choice to always project the weights on the $[0, 1]$ interval (recall Line 16 of Algorithm 2). Finally, in Section D.3 we provide the proof of Proposition 2.1 which was omitted from the main body of the paper.

### D.1 Statistical motivation

Recall the background on multiclass classification in Section 2.1. In this section we study hypothesis classes $\mathcal{F}$ and loss functions $\ell : \mathbb{R}^L \times \mathbb{R}^L :\to \mathbb{R}$ that are "well-behaved" with respect to a certain (standard in the machine learning literature) complexity measure we describe below.

For $\epsilon > 0$, a class $\mathcal{H}$ of functions $h : \mathcal{X} \to [0, 1]$ and an integer $n$, the "growth function" $\mathcal{N}_\infty(\epsilon, \mathcal{H}, n)$ is defined as

$$\mathcal{N}_\infty(\epsilon, \mathcal{H}, n) = \sup_{\mathbf{x} \in \mathcal{X}^n} \mathcal{N}(\epsilon, \mathcal{H}(\mathbf{x}), \|\cdot\|_\infty), \tag{6}$$

where $\mathcal{H}(\mathbf{x}) = \{(h(x_1), \ldots, h(x_n)) : h \in \mathcal{H}\} \subseteq \mathbb{R}^n$ and for $A \subseteq \mathbb{R}^n$ the number $\mathcal{N}(\epsilon, A, \|\cdot\|_\infty)$ is the smallest cardinality $A_0$ of a set $A_0 \subseteq A$ such that $A$ is contained in the union of $\epsilon$-balls centered at points in $A_0$, in the metric induced by $\|\cdot\|_\infty$ The growth number is a complexity measure of function classes commonly used in the machine learning literature [3, 17].

The following theorem from [30] provides large deviation bounds for function classes of polynomial growth.

**Theorem D.1** (Theorem 6, [30])**.** *Let $Z$ be a random variable taking values in $\mathcal{Z}$ distributed according to distribution $\mu$, and let $\mathcal{H} : \mathcal{Z} \to [0, 1]$ be a class of functions. Fix $\delta \in (0, 1), n \geq 16$ and set*

$$\mathcal{M}(n) = 10\mathcal{N}_\infty(1/n, \mathcal{H}, 2n).$$

*Then with probability at least $1 - \delta$ in the random vector $Z = (Z_1, \ldots, Z_n) \sim \mu^n$, for every $h \in \mathcal{H}$ we have:*

$$\left| \mathbb{E}[h(Z)] - \frac{1}{n} \sum_{i=1}^{n} h(Z_i) \right| \leq \sqrt{\frac{18 \mathbb{V}_n(h, Z) \ln(2\mathcal{M}(n)/\delta)}{n}} + \frac{15 \ln(2\mathcal{M}(n)/\delta)}{n-1},$$

*where $\mathbb{V}_n(h, Z)$ is the sample variance of the sequence $\{h(Z_i)\}_{i=1}^{n}$.*

A straightforward corollary of Theorem D.1 and Proposition 2.1, and the main motivation for our method, is the following corollary.

**Corollary D.2.** *Let $\ell : \mathbb{R}^L \times \mathbb{R}^L \to [0,1]$ be a loss function and fix $\delta > 0$. Consider any hypothesis class $\mathcal{F}$ of predictors $f : \mathcal{X} \to \mathbb{R}^L$, and the two induced classes $\mathcal{H} \subseteq [0,1]^{\mathbb{R}^L \times \mathbb{R}^L}$, $\mathcal{H}^w \subseteq [0,1]^{\mathbb{R}^L \times \mathbb{R}^L}$ of functions $h_f(x, y) := \ell(y, f(x))$ and $h_f^w(x, y) := w_f(x)\ell(y, f(x))$, respectively. Fix $\delta > 0$, $n \geq 16$, and set $\mathcal{M}(n) = 10\mathcal{N}_\infty(1/n, \mathcal{H}, 2n)$ and $\mathcal{M}^w(n) = 10\mathcal{N}_\infty(1/n, \mathcal{H}^w, 2n)$. Then, with probability at least $1 - \delta$ over $S = \{x_i, y_i\}_{i=1}^{n} \sim \mathbb{D}^n$,*

$$|R(f) + \mathrm{Bias}(f) - R_S(f)| \;\; = \;\; O\left( \sqrt{\mathbb{V}_S(f) \cdot \frac{\ln \frac{\mathcal{M}(n)}{\delta}}{n}} + \frac{\ln \frac{\mathcal{M}(n)}{\delta}}{n} \right) \tag{7}$$

$$|R(f) - R_S^w(f)| \;\; = \;\; O\left( \sqrt{\mathbb{V}_S^w(f) \cdot \frac{\ln \frac{\mathcal{M}^w(n)}{\delta}}{n}} + \frac{\ln \frac{\mathcal{M}^w(n)}{\delta}}{n} \right) \tag{8}$$

*where $\mathbb{V}_S(f), \mathbb{V}_S^w(f)$ are the sample variances of the loss values $\{h_f(x_i, y_i)\}_{i=1}^{n}$, $\{h_f^w(x_i, y_i)\}_{i=1}^{n}$, respectively.*

The following remark formally captures Theorem 4.1.

**Remark D.1.** *Under the assumptions of Corollary D.2, if we additionally have that $\mathcal{M}(n)$ and $\mathcal{M}^w(n)$ are polynomially bounded in $n$, then, for every $f \in \mathcal{F}$ it holds that*

$$\lim_{|S| \to \infty} R_S^w(f) = R(f) \quad \text{and} \quad \lim_{|S| \to \infty} R_S(f) = R(f) + \mathrm{Bias}(f).$$

## D.2 Studying the MSE of a fixed prediction

In this section we study the Mean-Squared-Error (MSE) of a fixed prediction $f(x)$ for an arbitrary instance $x \in \mathcal{X}$, predictor $f$, and loss function $\ell : \mathbb{R}^L \times \mathbb{R}^L \to \mathbb{R}_+$, in order to gain some understanding on when the importance weighting scheme could potentially underperform the standard unweighted approach from a bias-variance perspective (i.e., when the training sample is "small enough" so that asymptotic considerations are ill-suited). These considerations lead us to an additional justification for always projecting the weights to the $[0,1]$ interval (recall Line 16 of Algorithm 2).

Formally, we study the behavior of the quantities:

$$\mathrm{MSE}(x) \;\; = \;\; \mathbb{E}_{y|x} \left[ (\ell(f_{\text{true}}(x), f(x)) - \ell(y, f(x)))^2 \right],$$
$$\mathrm{MSE}^w(x) \;\; = \;\; \mathbb{E}_{y|x} [(\ell(f_{\text{true}}(x), f(x)) - w_f(x)\ell(y(x), f(x)))^2].$$

Recalling the definition of distortion (4) we have the following proposition.

**Proposition D.3.** *Let $\ell : \mathbb{R}^L \times \mathbb{R}^L \to \mathbb{R}_+$ be a bounded loss function. Fix $x \in \mathcal{X}$ and a predictor $f : \mathcal{X} \to \mathbb{R}^L$. We have $\mathrm{MSE}(x) < \mathrm{MSE}^w(x)$ if and only if:*

1. $\mathrm{distortion}_f(x) < 1/2$; *and*

2. $p(x) \in \left( 0, \frac{1 - 2 \cdot \mathrm{distortion}_f(x)}{(1 - \mathrm{distortion}_f(x))^2} \right)$.

*Proof sketch.* Via direct calculations we obtain:

$$\begin{aligned}
\mathrm{MSE}(x) \;\; &= \;\; \mathbb{E}_{y|x} \left[ (\ell(f_{\text{true}}(x), f(x)) - \ell(y, f(x)))^2 \right] \\
&= \;\; p(x)(\ell(f_{\text{true}}(x), f(x)) - \ell(y_{\text{adv}}(x), f(x)))^2 \\
&= \;\; p(x)\ell(f_{\text{true}}(x), f(x))^2 (1 - \mathrm{distortion}_f(x))^2
\end{aligned} \tag{9}$$

and

$$
\begin{aligned}
\mathrm{MSE}^w(x) & = \mathbb{E}_{y|x}[(\ell(f_{\mathrm{true}}(x), f(x)) - w_f(x)\ell(y, f(x)))^2] \\
& = (1 - p(x))\ell(f_{\mathrm{true}}(x), f(x))^2(1 - w_f(x))^2 \\
& \quad + p(x)(\ell(f_{\mathrm{true}}(x), f(x)) - w_f(x)\ell(y_{\mathrm{adv}}(x), f(x)))^2 \\
& = (1 - p(x))\ell(f_{\mathrm{true}}(x), f(x))^2(1 - w_f(x))^2 \\
& \quad + p(x)\ell(f_{\mathrm{true}}(x), f(x))^2(1 - w_f(x)\mathrm{distortion}_f(x))^2
\end{aligned}
\tag{10}
$$

Recalling the definition of weights (4), and combining it with (9) and (10) implies the claim.

$\square$

In words, Proposition D.3 implies that when the adversary does not have the power to corrupt the label of an instance $x$ with high enough probability, i.e., $p(x)$ is sufficiently small, and the prediction of the student is "close enough" to the adversarial label (i.e., when $\mathrm{distortion}_f(x)$ is small enough), then it potentially makes sense to use the unweighted estimator instead of the weighted one from a bias-variance trade-off perspective, as the former has smaller MSE in this case. Notice that this observation aligns well with our method as we always project $w_f(x)$ to $[0, 1]$ (observe that $w_f(x) > 1$ iff $\mathrm{distortion}_f(x) < 1$ and $p(x) > 0$ ).

### D.3    Proof of Proposition 2.1

Recall the weight, distortion and bias definitions in (4) and Proposition 2.1. We prove the first claim of Proposition 2.1 via direct calculations:

$$
\begin{aligned}
\mathbb{E}[R_S(f)] & = \mathbb{E}_{S \sim \mathbb{D}^n}\left[\frac{1}{n}\sum_{i=1}^{n}\ell(y_i, f(x_i))\right] \\
& = \mathbb{E}_{(x,y) \sim \mathbb{D}}[\ell(y, f(x))] \\
& = \mathbb{E}_{x \sim \mathbb{X}}[\mathbb{E}_{y|x}[\ell(y, f(x))]] \\
& = \mathbb{E}_{x \sim \mathbb{X}}[p(x)\ell(y_{\mathrm{adv}}(x), f(x)) + (1 - p(x))\ell(f_{\mathrm{true}}(x), f(x))] \\
& = \mathbb{E}_{x \sim \mathbb{X}}[\ell(f_{\mathrm{true}}(x), f(x))] + \mathbb{E}_{x \sim \mathbb{X}}[p(x) \cdot (\ell(y_{\mathrm{adv}}(x), f(x)) - \ell(f_{\mathrm{true}}, f(x)))] \\
& = \mathbb{E}_{x \sim \mathbb{X}}[\ell(f_{\mathrm{true}}(x), f(x))] + \mathbb{E}_{x \sim \mathbb{X}}\left[p(x) \cdot \left(\frac{\ell(y_{\mathrm{adv}}(x), f(x))}{\ell(f_{\mathrm{true}}(x), f(x))} - 1\right) \cdot \ell(f_{\mathrm{true}}(x), f(x))\right] \\
& = \mathbb{E}_{x \sim \mathbb{X}}[\ell(f_{\mathrm{true}}(x), f(x))] + \mathbb{E}_{x \sim \mathbb{X}}[p(x) \cdot (\mathrm{distortion}_f(x) - 1) \cdot \ell(f_{\mathrm{true}}(x), f(x))] \\
& = R(f) + \mathrm{Bias}(f).
\end{aligned}
$$

Similarly for the second claim:

$$
\begin{aligned}
\mathbb{E}[R_S^w(f)] & = \mathbb{E}_{S \sim \mathbb{D}^n}\left[\frac{1}{n}\sum_{i=1}^{n}w_f(x_i)\ell(y_i, f(x_i))\right] \\
& = \mathbb{E}_{x \sim \mathbb{X}}[\mathbb{E}_{y|x}[w_f(x)\ell(y, f(x))]] \\
& = \mathbb{E}_{x \sim \mathbb{X}}[w_f(x) \cdot (p(x)\ell(y_{\mathrm{adv}}(x), f(x)) + (1 - p(x))\ell(f_{\mathrm{true}}(x), f(x)))] \\
& = \mathbb{E}_{x \sim \mathbb{X}}\left[\frac{p(x)\ell(y_{\mathrm{adv}}(x), f(x)) + (1 - p(x))\ell(f_{\mathrm{true}}(x), f(x))}{1 + p(x) \cdot (\mathrm{distortion}_f(x) - 1)}\right] \\
& = \mathbb{E}_{x \sim \mathbb{X}}\left[\frac{\ell(f_{\mathrm{true}}(x), f(x)) + \ell(f_{\mathrm{true}}(x), f(x)) \cdot p(x) \cdot (\mathrm{distortion}_f(x) - 1)}{1 + p(x) \cdot (\mathrm{distortion}_f(x) - 1)}\right] \\
& = R(f),
\end{aligned}
\tag{11}
$$

concluding the proof.

## E    Extended theoretical motivation: optimization aspects

To prove our optimization guarantees, we analyze the reweighted objective in the fundamental case where the model $f(x; \Theta)$ is linear, i.e., $f(x; \Theta) = \Theta x \in \mathbb{R}^L$, and the loss $\ell(y, z)$ is convex in $z$ for

every $y$. In this case, the composition of the loss and the model $f(x; \Theta)$ is convex as a function of the parameter $\Theta \in \mathbb{R}^{L \times d}$. Recall that we denote by $f_{\text{true}}(x) : \mathbb{R}^d \mapsto \mathbb{R}^L$ the ground truth classifier and by $\mathbb{P}$ the "clean" distribution, i.e., a sample from $\mathbb{P}$ has the form $(x, f_{\text{true}}(x))$ where $x$ is drawn from a distribution $\mathbb{X}$ supported on (a subset of) $\mathbb{R}^d$. Finally, we denote by $\mathbb{D}$ the "noisy" labeled distribution on $\mathbb{R}^d \times \mathbb{R}^L$ and assume that the $x$-marginal of $\mathbb{D}$ is also $\mathbb{X}$.

**Notation** In what follows, for any elements $r, q$ of the same dimensions we denote by $r \cdot q$ their inner product. For example for two vectors $r, q \in \mathbb{R}^d$ we have $r \cdot q = \sum_{i=1}^d r_i q_i$. Similarly, for two matrices $\Theta, Q \in \mathbb{R}^{L \times d}$ we have $\Theta \cdot Q = \sum_{i=1}^L \sum_{j=1}^d \Theta_{ij} Q_{ij}$. We denote by $\| \cdot \|_2$ the $\ell_2$ for vectors and the spectral norm for matrices. We use $\otimes$ to denote the standard tensor (Kronecker) product between two vectors or matrices. For example, for two matrices $A, B$ we have $(A \otimes B)_{ijkl} = A_{ij} B_{kl}$ and for two vectors $v, u$ we have $(v \otimes u)_{ij} = v_i u_j$. We denote by $\| \cdot \|_F$ the Frobenious norm for matrices. We remark that we use standard asymptotic notation $O(\cdot)$, etc. and $\widetilde{O}(\cdot)$ to omit factors that are poly-logarithmic (in the appearing arguments).

For example, training a linear model $f(x; \Theta) = \Theta x$ with the Cross Entropy loss corresponds to using $\ell(t, y) = \sum_{i=1}^L t_i \log(\frac{e^{y_i}}{\sum_{j=1}^L e^{y_j}})$ and minimizing the objective

$$R(\Theta) = \mathbb{E}_{(x,y) \sim \mathbb{P}}[\ell(y, f(x; \Theta))] = \mathbb{E}_{(x,y) \sim \mathbb{P}}[\ell(y, \Theta x)].$$

More generally, in what follows we shall refer to the population loss over the clean distribution $\mathbb{P}$ as $R(\cdot)$, i.e.,

$$R(\Theta) \triangleq \mathbb{E}_{(x,y) \sim \mathbb{P}}[\ell(y, f(x; \Theta))].$$

We next give a general definition of debiasing weight functions, i.e., weighting mechanisms that make the corresponding objective function an unbiased estimator of the clean objective $R(\Theta)$ for every parameter vector $\Theta \in \mathbb{R}^d$.

**Definition E.1** (Debiasing Weights). *We say that a weight function $w(x, y_{\text{adv}}; \Theta) : \mathbb{R}^d \times \mathbb{R}^L \mapsto \mathbb{R}$ is a debiasing weight function if it holds that*

$$R^w(\Theta) \triangleq \mathbb{E}_{(x, y_{\text{adv}}) \sim \mathbb{D}}[w(x, y_{\text{adv}}; \Theta)\ell(y_{\text{adv}}, f(x, \Theta))] = R(\Theta).$$

**Remark E.1.** *We remark that the weight function $w(\cdot)$ depends on the current hypothesis, $\Theta$, and also on the noise advice $p(x)$ that we are given with every example. In order to keep the notation simple, we do not explicitly track these dependencies and simply write $w(x, y_{\text{adv}}; \Theta)$. We also remark that, in general, in order to construct the weight function $w$ we may also use "clean" data, which may be available, e.g., as a validation dataset, as we did in Section 2.2.*

Our main result is that, given a convex loss $\ell(\cdot)$ and a debiasing weight function $w(\cdot)$ that satisfy standard regularity assumptions, stochastic gradient descent on the reweighted objective produces a parameter vector with good generalization error. We first present the assumptions on the example distributions, the loss, and the weight function. In what follows, we view the gradient of a function $q(\Theta) : \mathbb{R}^{L \times d} \mapsto \mathbb{R}$ as an $L \times d$-matrix and the hessian $\nabla^2 q(\Theta)$ as an $(L \times d) \times (L \times d)$-tensor (or equivalently as a $dL \times dL$-matrix).

**Definition E.2** (Regularity Assumptions). *The $x$-marginal $\mathbb{X}$ of $\mathbb{D}$ and $\mathbb{P}$ is supported on (a subset of) the ball of radius $R > 0$, $\mathcal{B}_R \triangleq \{x \in \mathbb{R}^d : \|x\|_2 \leq R\}$.*

*The training model is linear $f(x; \Theta) = \Theta x$ and the parameter space is the unit ball, i.e., $\|\Theta\|_F \leq 1$.*

*For every label $y_{\text{adv}} \in \mathbb{R}^L$ in the support of $\mathbb{D}$, the loss $z \mapsto \ell(y_{\text{adv}}, z)$ is a twice differentiable, convex function in $z$. Moreover $\ell(y_{\text{adv}}, z)$ is $M_\ell$-bounded, $L_\ell$-Lipschitz, and $B_\ell$-smooth, i.e., $|\ell(y_{\text{adv}}, z)| \leq M_\ell$, $\|\nabla_z \ell(y_{\text{adv}}, z)\|_2 \leq L_\ell$, and $\|\nabla_z^2 \ell(y_{\text{adv}}, z)\|_2 \leq B_\ell$, for all $z$ with $\|z\|_2 \leq R$.*

*For every example $(x, y_{\text{adv}}) \in \mathbb{R}^d \times \mathbb{R}^L$ in the support of $\mathbb{D}$ the weight function $\Theta \mapsto w(x, y_{\text{adv}}; \Theta)$ is twice differentiable, $M_w$-bounded, $L_w$-Lipschitz, and $B_w$-smooth, i.e., $|w(x, y_{\text{adv}}; \Theta)| \leq M_w$, $\|\nabla_\Theta w(x, y_{\text{adv}}; \Theta)\|_F \leq L_w$, and $\|\nabla_\Theta^2 w(x, y_{\text{adv}}; \Theta)\|_2 \leq B_w$ for all $\Theta$ with $\|\Theta\|_F \leq 1$ [2].*

**Remark E.2.** *Observe that if a property in the above definition is satisfied by some parameter-value $Q$, then it is also satisfied for any other $Q' > Q$. For example, if the loss function is $0.5$-Lipschitz it is*

---

[2] Recall that, formally, $\nabla_\Theta^2 w(x, y_{\text{adv}}; \Theta)$ is a $(L \times d) \times (L \times d)$-tensor $G$. For this tensor $G$ we overload notation and set $\|G\|_2$ to be the standard $\ell_2$ operator norm when we view $G$ as an $(Ld) \times (Ld)$-matrix.

*also* 1-*Lipschitz. Therefore, to simplify the expressions, in what follows we shall assume (without loss of generality) that all the regularity parameters, i.e., $R, M_\ell, L_\ell, B_\ell, M_w, L_w, B_w$, are larger than* 1.

Since the loss is convex, it is straightforward to optimize the naive objective that does not reweight the loss and simply minimizes $\ell(\cdot)$ over the noisy examples, $R^{\mathrm{naive}}(\Theta) \triangleq \mathbb{E}_{(x,y_{\mathrm{adv}})\sim\mathbb{D}}[\ell(y_{\mathrm{adv}}, \Theta x)]$. We first show that (unsurprisingly) it is not hard to construct instances (even in binary classification) where optimizing the naive objective produces classifiers with large generalization error over clean examples. For simplicity, since in the following lemma we consider binary classification, we assume that the labels $y \in \{\pm 1\}$ and the parameter of the linear model is a vector $\theta \in \mathbb{R}^d$.

**Proposition E.3** (Naive Objective Fails). *Fix any* $c \in [0, 1]$. *Let* $\ell(\cdot)$ *be the Binary Cross Entropy loss, i.e.,* $\ell(t) = \log(1 + e^{-t})$. *There exists a "clean" distribution* $\mathbb{P}$ *and a noisy distribution* $\mathbb{D}$ *on* $\mathbb{R}^d \times \{\pm 1\}$ *so that the following hold.*

1. *The* $x$-*marginal of both* $\mathbb{P}$ *and* $\mathbb{D}$ *is uniform on a sphere.*

2. *The clean labels of* $\mathbb{P}$ *are consistent with a linear classifier* $\mathrm{sign}(\theta^* \cdot x)$.

3. $\mathbb{D}$ *has (total) label noise* $\mathrm{Pr}_{(x,y_{\mathrm{adv}})\sim\mathbb{D}}[y_{\mathrm{adv}} \neq \mathrm{sign}(\theta^* \cdot x)] = c \in [0, 1]$.

4. *The minimizer* $\widehat{\theta}$ *of the (population) naive objective* $R^{\mathrm{naive}}(\theta) = \mathbb{E}_{(x,y)\sim\mathbb{D}}[\ell(y_{\mathrm{adv}}\theta \cdot x)]$, *constrained on the unit has generalization error*

$$R(\widehat{\theta}) \geq \min_{\|\theta\|_2 \leq 1} R(\theta) + c/2 \,,$$

*where* $R(\theta)$ *is the "clean" risk,* $R(\theta) = \mathbb{E}_{(x,y)\sim\mathbb{P}}[\ell(y\theta \cdot x)]$.

Our positive results show that, having a debiasing weight function $w(\cdot)$ that is not very "wild" (see the regularity assumptions of Definition E.2) and optimizing the corresponding weighted objective $R^w(\Theta) = \mathbb{E}_{(x,y_{\mathrm{adv}})\sim\mathbb{D}}[w(x, y_{\mathrm{adv}}; \Theta)\ell(y_{\mathrm{adv}}, \Theta x)]$ with SGD, gives models with almost optimal generalization. The main issue with optimizing the reweighted objective is that, in general, we have no guarantees that the weight function preserves its convexity (recall that it depends on the parameter $\Theta$). However, we know that its population version corresponds to the clean objective $R(\cdot)$ which is a convex objective. We show that we can use the convexity of the underlying clean objective to show results for both single- and multi-pass stochastic gradient descent. We first focus on single-pass stochastic gradient descent where at every iteration a fresh noisy sample $(x, y_{\mathrm{adv}})$ is drawn from $\mathbb{D}$, see Algorithm 3.

---

**Algorithm 3** Single-Pass Stochastic Gradient Descent Algorithm

---

**Input:** Number of iterations $T$, Step size sequence $\eta^{(t)}$
**Output:** Parameter vector $\Theta^{(T)}$.

        Initialize $\Theta^{(1)} \leftarrow 0$.

        For $t = 1, \ldots, T$:

            Draw sample $(x^{(t)}, y_{\mathrm{adv}}^{(t)}) \sim \mathbb{D}$.
            Update using the gradient of the weighted objective:

$$\Theta^{(t+1)} \leftarrow \mathrm{proj}_{\mathcal{B}}\left(\Theta^{(t)} - \eta^{(t)}\nabla_\Theta\left(w(x^{(t)}, y_{\mathrm{adv}}^{(t)}; \Theta^{(t)})\, \ell(y_{\mathrm{adv}}^{(t)}, \Theta^{(t)}x^{(t)})\right)\right)$$

        Return $\Theta^{(T)}$.

---

**Theorem E.4** (Generalization of Reweighted Single-Pass SGD). *Assume that the example distributions* $\mathbb{P}, \mathbb{D}$, *the loss* $\ell(\cdot)$, *and the weight function* $w(\cdot)$ *satisfy the assumptions of Definition E.2. Set* $\kappa = L_w M_\ell + R M_w L_\ell$. *After* $T = \Omega(\kappa^2/\epsilon^2)$ *SGD iterations (with a step size sequence that depends on the regularity parameters of Definition E.2, see Algorithm 3), with probability at least* 99%, *it holds*

$$R(\Theta^{(T)}) \leq \min_{\|\Theta\|_F \leq 1} R(\Theta) + \epsilon \,.$$

The main observation in the single-pass setting is that, since the weight function $w(\cdot)$ is debiasing, we can view the gradients of the reweighted objective as stochastic gradients of the true objective over the clean samples. Therefore, as long as we draw a fresh i.i.d. noisy sample $(x, y) \sim \mathbb{D}$ at each round, the corresponding sequence of gradients corresponds to stochastic unbiased estimates of the gradients of the true loss $R(\Theta)$. We next turn our attention to multi-pass SGD (see Algorithm 4), where at each round we pick one of the $N$ samples with replacement and update according to its gradient. The key difference between single- and multi-pass SGD is that the expected loss over the stochastic algorithm for single-pass SGD is the population risk, while the expected loss for multi-pass SGD is the empirical risk. In other words, in the multi-pass setting we have a stochastic gradient oracle to the empirical reweighted objective $\widehat{R}^w(\Theta) = \frac{1}{N} \sum_{i=1}^{N} w(x^{(i)}, y_{\text{adv}}^{(i)}; \Theta) \, \ell(y_{\text{adv}}^{(i)}, \Theta x^{(i)})$, which is not necessarily convex. Our second theorem shows that under the regularity conditions of Definition E.2 multi-pass SGD also achieves good generalization error.

**Theorem E.5** (Generalization of Reweighted Multi-Pass SGD). *Assume that the example distributions $\mathbb{P}, \mathbb{D}$, the loss $\ell(\cdot)$, and the weight function $w(\cdot)$ satisfy the assumptions of Definition E.2. Set $\kappa = R M_\ell L_\ell B_\ell M_w L_w B_w$ and define the empirical reweighted objective with $N = (dL)^2/\epsilon^2 \operatorname{poly}(\kappa)$ i.i.d. samples $(x^{(1)}, y_{\text{adv}}^{(1)}), \dots, (x^{(N)}, y_{\text{adv}}^{(N)})$ from the noisy distribution $\mathbb{D}$ as*

$$\widehat{R}^w(\Theta) = \frac{1}{N} \sum_{i=1}^{N} w(x^{(i)}, y_{\text{adv}}^{(i)}; \Theta) \, \ell(y_{\text{adv}}^{(i)}, \Theta x^{(i)}) \,.$$

*Then, after $T = \operatorname{poly}(\kappa)/\epsilon^4$ iterations, multi-pass SGD with constant step size sequence $\eta^{(t)} = C/\sqrt{T}$ [3] (see Algorithm 4) on $\widehat{R}^w(\cdot)$ outputs a list $\Theta^{(1)}, \dots, \Theta^{(T)}$ that, with probability at least 99%, contains a vector $\widehat{\Theta}$ that satisfies*

$$R(\widehat{\Theta}) \leq \min_{\|\Theta\|_F \leq 1} R(\Theta) + \epsilon \,.$$

We remark that our analysis also applies to the multi-pass SGD variant where, at every epoch we pick a random permutation of the $N$ samples and update with their gradients sequentially.

---

**Algorithm 4** Multi-Pass Stochastic Gradient Descent Algorithm

---

**Input:** Number of Rounds $T$, Number of Samples $N$, Step size sequence $\eta^{(t)}$.
**Output:** List of weight vectors $\Theta^{(1)}, \dots, \Theta^{(T)}$.

    Draw $N$ i.i.d. samples $(x^{(1)}, y_{\text{adv}}^{(1)}), \dots, (x^{(N)}, y_{\text{adv}}^{(N)}) \sim \mathbb{D}$.

    Initialize $\Theta^{(1)} \leftarrow 0$.

    For $t = 1, \dots, T$:

        Pick $I$ uniformly at random from $\{1, \dots, N\}$ and update using the gradient of the reweighted objective:

$$\Theta^{(t+1)} \leftarrow \operatorname{proj}_{\mathcal{B}} \left( \Theta^{(t)} - \eta^{(t)} \nabla_\Theta \left( w(x^{(I)}, y_{\text{adv}}^{(I)}; \Theta^{(t)}) \ell(y_{\text{adv}}^{(I)}, \Theta^{(t)} x^{(I)}) \right) \right) \,.$$

    Return $\Theta^{(1)}, \dots, \Theta^{(T)}$.

---

## E.1 The proof of Proposition E.3

In this subsection we restate and prove Proposition E.3.

**Proposition E.6** (Naive Objective Fails (Restate of E.3) ). *Fix any $c \in [0, 1]$. Let $\ell(\cdot)$ be the Binary Cross Entropy loss, i.e., $\ell(t) = \log(1 + e^{-t})$. There exists a "clean" distribution $\mathbb{P}$ and a noisy distribution $\mathbb{D}$ on $\mathbb{R}^d \times \{\pm 1\}$ so that the following hold.*

    *1. The $x$-marginal of both $\mathbb{P}$ and $\mathbb{D}$ is uniform on a sphere.*

---

[3] $C$ is a constant that depends on the regularity parameters of Definition E.2.

2. *The clean labels of $\mathbb{P}$ are consistent with a linear classifier $\mathrm{sign}(\theta^* \cdot x)$.*

3. *$\mathbb{D}$ has (total) label noise $\mathrm{Pr}_{(x, y_{\mathrm{adv}}) \sim \mathbb{D}}[y_{\mathrm{adv}} \neq \mathrm{sign}(\theta^* \cdot x)] = c \in [0, 1]$.*

4. *The minimizer $\widehat{\theta}$ of the (population) naive objective $R^{\mathrm{naive}}(\theta) = \mathbb{E}_{(x,y) \sim \mathbb{D}}[\ell(y_{\mathrm{adv}} \theta \cdot x)]$, constrained on the unit has generalization error*

$$R(\widehat{\theta}) \geq \min_{\|\theta\|_2 \leq 1} R(\theta) + c/2\,,$$

*where $R(\theta)$ is the "clean" risk, $R(\theta) = \mathbb{E}_{(x,z) \sim \mathbb{P}}[\ell(z\theta \cdot x)]$.*

*Proof.* We set the $\mathbb{X}$-marginal to be the uniform distribution on a sphere of radius $R > 0$ to be specified later in the proof. We first observe that the unit vector $\theta^*$ minimizes the (clean) Binary Cross Entropy $R(\theta)$. We can now pick a different parameter vector $\widetilde{\theta}$ with angle $\phi \in [0, \pi]$ with $\theta^*$, and construct a noisy instance as follows: we first draw $x \sim \mathbb{X}$ (recall that we want the $x$-marginal of the noisy distribution to be the same as the "clean") and then set $y_{\mathrm{adv}} = \mathrm{sign}(\widetilde{\theta} \cdot x)$. By the symmetry of the uniform distribution on the sphere we have that

$$\mathbf{Pr}_{(x,y) \sim \mathbb{D}}[y_{\mathrm{adv}} \neq \mathrm{sign}(\theta^* \cdot x)] = \mathbf{Pr}_{x \sim \mathbb{D}_x}[\mathrm{sign}(\widetilde{\theta} \cdot x) \neq \mathrm{sign}(\theta^* \cdot x)] = \frac{\phi}{\pi}\,.$$

Therefore, by picking the angle $\phi$ to be equal to $\pi c$ we obtain that $\mathbf{Pr}_{x \sim \mathbb{D}_x}[\mathrm{sign}(\widetilde{\theta} \cdot x) \neq \mathrm{sign}(\theta^* \cdot x)] = c$ as required by Proposition E.3. Moroever, we have that the minimizer of the "naive" BCE objective (constrained on the unit ball) is $\widetilde{\theta}$ and the minimizer of the clean objective is $\theta^*$. We have that

$$R(\widetilde{\theta}) = \mathbb{E}_{x \sim \mathbb{X}}[\log(1 + e^{-\mathrm{sign}(\theta^* \cdot x)\widetilde{\theta} \cdot x})] \geq \mathbb{E}_{x \sim \mathbb{X}}[\mathbb{1}\{\mathrm{sign}(\theta^* \cdot x)\widetilde{\theta} \cdot x < 0\}]$$
$$= \mathbf{Pr}_{x \sim \mathbb{X}}[\mathrm{sign}(\widetilde{\theta} \cdot x) \neq \mathrm{sign}(\theta^* \cdot x)] = c\,.$$

Moreover, we have that

$$R(\theta^*) = \mathbb{E}_{x \sim \mathbb{X}}[\log(1 + e^{-\mathrm{sign}(\theta^* \cdot x)\theta^* \cdot x})]$$
$$= \mathbb{E}_{x \sim \mathbb{X}}[\log(1 + e^{-|\theta^* \cdot x|})]$$

We next need to bound from below the "margin" of the optimal weight vector $\theta^*$, i.e., provide a lower bound on $|\theta^* \cdot x|$ that holds with high probability. We will use the following anti-concentration inequality on the probability of a origin-centered slice under the uniform distribution on the sphere. For a proof see, e.g., Lemma 4 in [10].

**Lemma E.7** (Anti-Concentration of Uniform vectors, [10]). *Let $v \in \mathbb{R}^d$ be any unit vector and let $\mathbb{X}$ be the uniform distribution on the sphere. It holds that*

$$\mathbf{Pr}_{x \sim \mathbb{X}}\left[|v \cdot x| \leq \frac{\gamma}{\sqrt{d}}\right] \leq \gamma\,.$$

Using Lemma E.7, we obtain that

$$\mathbb{E}_{x \sim \mathbb{X}}[\log(1 + e^{-|\theta^* \cdot x|})] \leq \log(2)\gamma + \log(1 + e^{-\gamma R/\sqrt{d}})(1 - \gamma) \leq 2\gamma + e^{-\gamma R/\sqrt{d}}\,,$$

where, at the last step, we used the elementary inequality $\log(1 + x) \leq x$. Assuming that $R/\sqrt{d}$ is much larger than 1, we can pick $\gamma = (\sqrt{d}/R)\log(R/\sqrt{d})$. For this choice of $\gamma$ we obtain that $\mathbb{E}_{x \sim \mathbb{X}}[\log(1 + e^{-|\theta^* \cdot x|})] = O(\sqrt{d}/R\log(d/R))$. Therefore, for $R = O(\sqrt{d}/c)$ we obtain that

$$\mathbb{E}_{x \sim \mathbb{X}}[\log(1 + e^{-|\theta^* \cdot x|})] \leq c/2\,.$$

Therefore, combining the above bounds we obtain that $R(\widehat{\theta}) - R(\theta^*) \geq c - c/2 \geq c/2$. $\qquad \square$

## E.2 The Proof of Theorem E.4

In this section we restate and prove our result on the generalization error of single-pass stochastic gradient descent on the weighted objective.

**Theorem E.8** (Generalization of Reweighted Single-Pass SGD (Restate of E.4)). *Assume that the example distributions $\mathbb{P}, \mathbb{D}$ and the $\ell(\cdot)$ and weight function $w(\cdot)$ satisfy the assumptions of Definition E.2. Set $\kappa = L_w M_\ell + R M_w L_\ell$. After $T = \Omega(\kappa^2/\epsilon^2)$ SGD iterations (see Algorithm 3), with probability at least $99\%$, it holds*

$$R(\Theta^{(T)}) \leq \min_{\|\Theta\|_F \leq 1} R(\Theta) + \epsilon.$$

*Proof.* We observe that, since $w$ is a debiasing weight function, given a sample $(x^{(t)}, y^{(t)}_{\mathrm{adv}}) \sim \mathbb{D}$ it holds that $\nabla(w(x^{(t)}, y^{(t)}_{\mathrm{adv}}; \Theta)\ell(y_{\mathrm{adv}}, \Theta x^{(t)}))$ is an unbiased gradient estimate of $\nabla_\Theta R(\Theta)$. We will use the following result on the convergence of the last-iterate of SGD for convex objectives. For simplicity, we state the following theorem for the case where the parameter $\theta$ is a vector in $\mathbb{R}^d$ (instead of a $L \times d$ matrix).

**Lemma E.9** (Last Iterate Stochastic Gradient Descent [22]). *Let $\mathcal{W}$ be a closed convex set of diameter $R$. Moreover, let $F : \mathbb{R}^d \mapsto \mathbb{R}$ be a convex, $L$-Lipschitz function. Define the stochastic gradient descent iteration as*

$$\theta^{(0)} \leftarrow 0$$

$$\theta^{(t+1)} \leftarrow \mathrm{proj}_{\mathcal{W}}\left(\theta^{(t)} - \eta^{(t)} g^{(t)}(\theta^{(t)})\right)$$

*where $g^{(t)}(\theta^{(t)})$ is an unbiased gradient estimate of $\nabla_\theta f_{\mathrm{true}}(\theta^{(t)})$. Assume that for all $t \in [T]$ it holds $\|g^{(t)}(\theta)\|_2 \leq L$ for all $\theta \in \mathcal{W}$. There exists a step size sequence $\eta^{(t)}$ that depends only on $T, L, R$ such that, with probability at least $1 - \delta$, it holds*

$$F(\theta^{(T)}) \leq F(\theta^*) + O\left(RL\sqrt{\frac{\log(1/\delta)}{T}}\right).$$

To simplify notation we let $\ell^{(t)}(\Theta) \triangleq \ell(y^{(t)}_{\mathrm{adv}}, \Theta x^{(t)})$ and $w^{(t)}(\Theta) \triangleq w(x^{(t)}, y^{(t)}_{\mathrm{adv}}; \Theta)$. For a sample $(x^{(t)}, y^{(t)}_{\mathrm{adv}})$, the gradient $g^{(t)}$ of the weighted loss is:

$$g^{(t)} = \nabla_\Theta(w^{(t)}(\Theta)\ell^{(t)}(\Theta))$$

$$= \ell^{(t)}(\Theta)\nabla_\Theta w^{(t)}(\Theta) + w^{(t)}(\Theta)\nabla_\Theta \ell^{(t)}(\Theta)$$

$$= \ell^{(t)}(\Theta)\nabla_\Theta w^{(t)}(\Theta) + w^{(t)}(\Theta)\nabla_z \ell(x^{(t)} y^{(t)}_{\mathrm{adv}}, z)\big|_{z=\Theta x^{(t)}} (x^{(t)})^T \in \mathbb{R}^{L \times d}.$$

Using the triangle inequality for the Frobenious norm and the assumptions of Definition E.2 on the functions $w(\cdot)$ and $\ell(\cdot)$, we obtain that $\|g^{(t)}\|_F \leq L_w M_\ell + R M_w L_\ell$. Using Lemma E.9 we obtain that with $T = \Omega((L_w M_\ell + R M_w L_\ell)^2/\epsilon^2)$, the last iteration of Algorithm 3 satisfies the claimed guarantee. $\qquad \square$

## E.3 The proof of Theorem E.5

In this section we prove our result on multi-pass SGD. For convenience, we first restate it.

**Theorem E.10** (Generalization of Multi-Pass SGD (Restate of E.5)). *Set $\kappa = R M_\ell L_\ell B_\ell M_w L_w B_w$ and define the empirical reweighted objective with $N = d^2/\epsilon^2 \, \mathrm{poly}(\kappa)$ i.i.d. samples $(x^{(1)}, y^{(1)}_{\mathrm{adv}}), \ldots, (x^{(N)}, y^{(N)}_{\mathrm{adv}})$ from the noisy distribution $\mathbb{D}$ as*

$$\widehat{R}^w(\Theta) = \frac{1}{N}\sum_{i=1}^{N} w(x^{(i)}, y^{(i)}_{\mathrm{adv}}; \Theta)\, \ell(\theta \cdot x^{(i)} y^{(i)}_{\mathrm{adv}}).$$

*Then, after $T = \mathrm{poly}(\kappa)/\epsilon^4$ iterations, multi-pass SGD (see Algorithm 4) on $\widehat{R}^w(\cdot)$ outputs a list $\theta^{(1)}, \ldots, \theta^{(T)}$ that, with probability at least $99\%$, contains a vector $\widehat{\theta}$ that satisfies*

$$R(\widehat{\Theta}) \leq \min_{\|\Theta\|_F \leq 1} R(\Theta) + \epsilon.$$

*Proof.* To prove the theorem we shall first show that all stationary points of the empirical objective (which for arbitrary weight functions $w(\cdot)$ may be non-convex) will have good generalization guarantees. Before we proceed we formally define approximate stationary points. To simplify notation we shall assume that the parameter is a vector $\theta \in \mathbb{R}^d$. The definition extends directly to the case where $L$ is a function of a parameter matrix $\Theta$ by using the corresponding matrix inner product.

**Definition E.11** ($\epsilon$-approximate Stationary Points). *Let $L : \mathbb{R}^d \mapsto \mathbb{R}$ be a differentiable function and $C$ be any convex subset of $\mathbb{R}^d$. A vector $\theta \in \mathbb{R}^d$ is an $\epsilon$-approximate stationary point of $L(\cdot)$ if for every $\theta' \in C$ it holds that*

$$\left| \nabla_\theta L(\theta) \cdot \frac{\theta' - \theta}{\|\theta' - \theta\|_2} \right| \le \epsilon \,.$$

**Proposition E.12.** *Set $\kappa = R M_\ell L_\ell B_\ell M_w L_w B_w$ and define the empirical reweighted objective with $N = \widetilde{O}((dL/\epsilon)^2) \operatorname{poly}(\kappa) \log(1/\delta)$ i.i.d. samples $(x^{(1)}, y^{(1)}), \ldots, (x^{(N)}, y^{(N)})$ from the noisy distribution $\mathbb{D}$ as*

$$\widehat{R}^w(\theta) = \frac{1}{N} \sum_{i=1}^N w(x^{(i)}, y^{(i)}; \Theta)\, \ell(y^{(i)}, \Theta x^{(i)}) \,.$$

*Let $\widehat{\Theta}$ be any $\epsilon$-stationary point of $\widehat{R}^w(\Theta)$ constrained on $\mathcal{B}_R$. Then, with probability at least $1 - \delta$, it holds that*

$$R(\widehat{\Theta}) \le \min_{\|\Theta\|_F \le 1} R(\Theta) + \epsilon \,.$$

*Proof.* We first show that, as long as the empirical gradients are close to the population gradients, any stationary point of the weighted empirical objective will achieve good generalization error. In what follows we shall denote by $\Theta^*$ the parameter that minimizes the clean objective:

$$\Theta^* \triangleq \operatorname*{arg\,min}_{\|\Theta\|_F \le 1} R(\Theta) \,.$$

Since the population objective is convex in $\Theta$, we have that for any $\Theta$ it holds that

$$
\begin{aligned}
R(\Theta) - R(\Theta^*) &\le \nabla_\Theta R(\Theta) \cdot (\Theta - \Theta^*) \\
&= (\nabla_\Theta R(\Theta) - \nabla_\Theta \widetilde{\mathcal{L}}^w(\Theta)) \cdot (\Theta - \Theta^*) + \nabla_\Theta \widehat{R}^w(\theta) \cdot (\Theta - \Theta^*) \\
&\le 2\|\nabla_\Theta \widehat{R}^w(\Theta) - \nabla_\Theta \widehat{R}^w(\Theta)\|_2 + \nabla_\Theta \widehat{R}^w(\Theta) \cdot (\Theta - \Theta^*) \,.
\end{aligned}
$$

We have that the contstraint set $\|\Theta\|_F \le 1$ is convex and therefore for a stationary point $\widehat{\Theta}$ of $\mathcal{L}^w(\Theta)$ we have that $|\nabla_\theta \widehat{R}^w(\Theta) \cdot (\Theta - \Theta^*)| \le \epsilon \|\Theta - \Theta^*\|_F \le 2\epsilon$. Therefore, $\widehat{\Theta}$ satisfies

$$R(\widehat{\Theta}) - R(\Theta^*) \le 2\|\nabla_\Theta R(\widehat{\Theta}) - \nabla_\Theta \widehat{R}^w(\widehat{\Theta})\|_2 + 2\epsilon \,.$$

Since $w(\cdot)$ is a debiasing weighting function, we know that, as the number of samples $N \to \infty$, the empirical gradients of the reweighted objective will converge to the gradients of the population clean objective $R(\cdot)$, i.e., it holds that $\nabla_\theta \widehat{R}^w(\Theta) \to \nabla_\Theta R(\Theta)$. Therefore, to finish the proof, we need to provide a uniform convergence bound for the gradient field of the empirical objective. We first consider estimating the gradient of some fixed parameter matrix $\Theta$. We will use McDiarmid's inequality.

**Lemma E.13** (McDiarmid's Inequality). *Let $x_1, \ldots, x_n$ be $n$ i.i.d. random variables taking values in $\mathcal{X}$. Let $\phi : \mathcal{X}^n \to \mathbb{R}$ be such that $|\phi(x) - \phi(x')| \le b_i$ whenever $x$ and $x'$ differ only on the $i$-th coordinate. It holds that*

$$\mathbf{Pr}\left[ |\phi(x_1, \ldots, x_n) - \mathbb{E}[\phi(x_1, \ldots, x_n)]| \ge \epsilon \right] \le 2 \exp\left( -\frac{2\epsilon^2}{\sum_{i=1}^n b_i^2} \right)$$

We consider the $n$ i.i.d. random variables $(x^{(t)}, y_{\mathrm{adv}}^{(t)})$. We have that the empirical gradient of the weighted loss function is equal to

$$\widehat{g}(\Theta) \triangleq \frac{1}{N} \sum_{t=1}^N \left( \ell(y_{\mathrm{adv}}; \Theta x^{(t)})\, \nabla_\Theta w(x^{(t)}, y_{\mathrm{adv}}^{(t)}; \Theta) + w(x^{(t)}, y_{\mathrm{adv}}^{(t)}; \Theta) \nabla \ell(y_{\mathrm{adv}}^{(t)}, \Theta x^{(t)})\, (x^{(t)})^T \right) \,.$$

We have that $w(x^{(t)}, y_{\text{adv}}^{(t)}; \Theta)$ is $M_w$-bounded and $L_w$-Lipschitz, $\ell$ is $M_\ell$-bounded and $L_\ell$-Lipschitz, and $\|x^{(t)}\|_2 \leq R$. Therefore, the maximum value of each coordinate of each term in the sum of the empirical gradient $\widehat{g}(\theta)$ is bounded by $L_q \triangleq L_w M_\ell + R M_w L_\ell$. Using this fact we obtain that each coordinate of the empirical gradient is a function of the $N$ i.i.d. random variables $(x^{(t)}, y_{\text{adv}}^{(t)})$ that satisfies the bounded differences assumption with constants $b_1, \ldots, b_N$ that satisfy $b_t \leq L_q/N$. From Lemma E.13, we obtain that

$$\mathbf{Pr}\left[\|\widehat{g}(\Theta) - \nabla_\Theta R(\Theta)\|_F \geq \epsilon\right] \leq \sum_{i=1}^d \sum_{j=1}^L \mathbf{Pr}\left[|(\widehat{g}(\Theta))_{ij} - (\nabla_\Theta R(\Theta))_{ij}| \geq \epsilon/\sqrt{dL}\right]$$

$$\leq 2dL \exp\left(-\Omega\left(N\epsilon^2/(dL\, L_q^2)\right)\right)^4.$$

We next need to provide a uniform convergence guarantee over the whole parameter space $\|\Theta\|_F \leq 1$. We will use the following standard lemma bounding the cardinality of an $\epsilon$-net of the unit ball in $d$-dimensions. For a proof see, e.g., [46].

**Lemma E.14** (Cover of the Unit Ball [46]). *Let $\mathcal{B}$ be the $d$-dimensional unit ball around the origin. There exists an $\epsilon$-net of $\mathcal{B}$ with cardinality at most $(1 + 2/\epsilon)^d$.*

Since we plan to construct a net for the gradient of $w(x, y_{\text{adv}}; \Theta)\ell(y_{\text{adv}}; \Theta x)$ we first need to show that the weighted loss $w(x, y_{\text{adv}}; \Theta)\ell(y_{\text{adv}}; \Theta x)$ is a smooth function of its parameter $\Theta$ or, in other words, that its gradients do not change very fast with respect to $\Theta$. The following lemma follows directly from the regularity assumptions of Defintion E.2 and the chain and product rules for the derivatives.

**Lemma E.15.** *For all $(x, y) \in \mathcal{B}_R \times \mathbb{R}^L$, it holds that the function $q(\Theta) = w(x, y; \Theta)\ell(y; \Theta x)$ is $B_q$-smooth for all $\Theta$ with $\|\Theta\|_F \leq 1$, with $B_q = M_\ell B_w + 2L_\ell L_w R + M_w B_\ell R^2$.*

*Proof.* For simplicy we shall denote $\nabla_z \ell(y; z)$ simply by $\nabla \ell(y; z)$ and similarly $\nabla_z^2 \ell(y; z)$ by $\nabla^2 \ell(y; z)$. Using the chain rule, we have that the gradient of the weighted loss $q(\Theta)$ is equal to

$$\nabla_\Theta q(\Theta) = \nabla_\Theta w(x, y; \Theta)\, \ell(y; \Theta x) + w(x, y; \Theta)\nabla\ell(y; \Theta x)x^T.$$

Using again the chain and product rules we find the Hessian of $q(\Theta)$:

$$\nabla_\Theta^2 q(\Theta) = \nabla_\Theta^2 w(x, y; \Theta)\, \ell(y; \Theta x) + (\nabla\ell(y; \Theta x)x^T) \otimes \nabla_\Theta w(x, y; \Theta)$$
$$+ \nabla_\Theta w(x, y; \Theta) \otimes (\nabla\ell(y; \Theta x)x^T) + w(x, y; \Theta)H,$$

where $H$ is the $(L \times d) \times (L \times d)$ tensor with element $H_{ijkl} = \nabla^2 \ell(y; \Theta x)_{ik} x_j x_l$. Recall that we view $\nabla_\Theta^2 q(\Theta)$ as an $Ld \times Ld$ and to prove that it is smooth we have to find its operator (spectral) norm. Using the assumptions of Definition E.2 we obtain that $\|\nabla_\Theta^2 w(x, y; \Theta)\|_2 \leq B_w M_\ell$. For the term $(\nabla\ell(y; \Theta x)x^T) \otimes \nabla_\Theta w(x, y; \Theta)$ we consider any $q \in \mathbb{R}^{Ld}$ with $\|q\|_2 = 1$. We assume that $q$ is indexed as $q_{ij}$ for $i = 1, \ldots, L$ and $j = 1, \ldots, d$. We have

$$q^T((\nabla\ell(y; \Theta x)x^T) \otimes \nabla_\Theta w(x, y; \Theta))q = \left(\sum_{ij} q_{ij}(\nabla\ell(y; \Theta x))_i x_j\right)\left(\sum_{kl} q_{kl}(\nabla_\Theta w(x, y; \Theta))_{kl}\right)$$
$$\leq RL_\ell L_w.$$

Similarly, we bound the spectral norm of the term $\nabla_\Theta w(x, y; \Theta) \otimes (\nabla\ell(y; \Theta x)x^T)$. Finally for the term $H$ we have

$$q^T H q = \sum_{ijkl} q_{ij} x_j (\nabla\ell(y; \Theta x))_{ik} q_{kl} x_l = \sum_{ik} s_i (\nabla\ell(y; \Theta x))_{ik} s_k,$$

where $s \in \mathbb{R}^L$ has $s_i = \sum_j q_{ij} x_j$. Observe that since $\|x\|_2 \leq R$ and $\|q\|_2 = 1$ we have that $\|s\|_2 \leq R$. Therefore, from the assumption of Definition E.2, we obtain that $\|H\|_2 \leq R^2 B_\ell$.

We conclude that the function $q(\theta)$ is $B_q$-smooth on the unit ball $\mathcal{B}$ with $B_q = M_\ell B_w + 2L_\ell L_w R + M_w B_\ell R^2$. $\qquad\square$

Let $\mathcal{N}_\epsilon$ be an $\epsilon$-net of the unit ball $\mathcal{B}$. Using Lemma E.15 we first observe that the vector maps $\theta \mapsto \widetilde{g}(\Theta)$ and $\Theta \mapsto \nabla_\Theta R(\Theta)$ are both $B_q$-Lipschitz, where $B_q$ is the constant defined in Lemma E.15. Using the triangle inequality and the fact that $\widehat{g}(\cdot)$ and $\nabla_\Theta R(\cdot)$ are $B_q$-Lipschitz, we have that

$$\max_{\|\Theta\|_F \leq 1} \|\widehat{g}(\Theta) - \nabla_\Theta R(\Theta)\|_2 \leq 2B_q\epsilon + \max_{\Theta \in \mathcal{N}_\epsilon} \|\widehat{g}(\Theta) - \nabla_\Theta R(\Theta)\|_2 .$$

Combining the above, and performing a union bound over the $\epsilon$-net $\mathcal{N}_\epsilon$, we obtain that

$$\mathbf{Pr}\left[ \max_{\|\Theta\|_F \leq 1} \|\widetilde{g}(\Theta) - \nabla_\Theta R(\Theta)\|_F \geq (2B_q + 1)\epsilon \right] \leq (1 + 2/\epsilon)^{dL} \exp\left( -\Omega\left( N\epsilon^2/(dL\, L_q^2) \right) \right) .$$

We conclude that with $N = \widetilde{\Omega}((dL)^2 L_q^2 B_q^2/\epsilon^2 \log(1/\delta))$ samples, it holds that $\|\widehat{g}(\Theta) - \nabla_\Theta R(\Theta)\|_2 \leq \epsilon$, uniformly for all parameters $\Theta$ with $\|\Theta\|_F \leq 1$, with probability at least $1 - \delta$. $\quad\square$

We now have to show that the multi-pass SGD finds an approximate stationary point of the empirical objective. We will use the following result on non-convex projected SGD. To simplify notation, we state the following optimization lemma assuming that the parameter is a vector $\theta \in \mathbb{R}^d$.

**Lemma E.16** (Non-Convex Projected Stochastic Gradient Descent [11]). *Let $\mathcal{W}$ be a closed convex set of diameter $R$. Moreover, let $F : \mathbb{R}^d \mapsto \mathbb{R}$ be an $L$-Lipschitz and $B$-smooth function. Define the stochastic gradient descent iteration as*

$$\theta^{(0)} \leftarrow 0$$
$$\theta^{(t+1)} \leftarrow \mathrm{proj}_{\mathcal{W}}\left( \theta^{(t)} - \eta^{(t)} g^{(t)}(\theta^{(t)}) \right)$$

*where $g^{(t)}(\theta^{(t)})$ is an unbiased gradient estimate of $\nabla_\theta F(\theta^{(t)})$. Fix a number of iterations $T \geq 1$ and assume that for all $t \in [T]$ it holds $\|g^{(t)}(\theta)\|_2 \leq L$ for all $\theta \in \mathcal{W}$. Set the step-size $\eta^{(t)} = \Theta(\sqrt{R/(BL^2T)})$. With probability at least $99\%$, there exists a $t \in \{1, \ldots, T\}$ such that $\theta^{(t)}$ is an $O\left( \frac{\sqrt{BLR}}{T^{1/4}} \right)$-stationary point of $F(\cdot)$ constrained on $\mathcal{W}$.*

Theorem E.5 now follows directly by applying Lemma E.16 on the empirical objective to find an $\epsilon$-approximate stationary point and then using Proposition E.12.

$\square$