# OpenReview forum: "Weighted Distillation with Unlabeled Examples"
_NeurIPS.cc/2022/Conference — NeurIPS 2022 Accept_

### Official Review · Reviewer_HUb8 · 2022-07-06

**Rating:** 5
**Confidence:** 2
**Soundness:** 2 fair
**Presentation:** 3 good
**Contribution:** 2 fair

**Summary:**

In this paper, the authors proposed a reweighting method to improve the performance of data distillation on the unlabeled dataset. Experiments on various datasets (SVHN, CIFAR-10/100, CelebA, and ImageNet) demonstrate the performance of the proposed method.

**Questions:**

I'm wondering why or how likely the pseudo labels are not reliable in the considered setting of the paper. In many cases in distillation with unlabeled data, the unlabeled dataset is different from the labeled data (or comes from a different distribution) used to train the teacher model. Then, it is likely that the pseudo labels given from the teacher model are unreliable. But when the labeled data and the unlabeled data come from the same distribution (i.e., the setting used in the experiments), this may be not a big issue.

**Limitations:**

It seems that the authors haven't addressed the potential negative societal impact of their work.

**Strengths And Weaknesses:**

Advantages:
1. The research topic, distillation with unlabeled examples, is very close to many real-world applications.
2. The proposed method has some provable guarantees.

Disadvantages:
1. The proposed method, especially the procedure for weight estimation is quite complex and computationally expensive. If the labels from the teacher model are not reliable (can be regarded as label noise), why not apply some label noise methods to tackle this problem?
2. In the experiments, the performance gain is not that much (e.g., on CelebA and ImageNet) when compared with the unweighted method.
3. The paper does not show how and why using K-nn to learn the mapping between label confidence and the estimates of p(x) and the distortion(x). Is there any other method that can be used?

---

> ### Author Response · Authors · 2022-08-02
> **Response to reviewer HUb8**
>
>
> We thank the reviewer for their constructive feedback.
>
> - *... is quite complex and computationally expensive*
>
> 	**Our method is easy to implement, it is hyperparameter-free, and computationally inexpensive**. Indeed, the time for computing the weight for each example is insignificant compared to the actual training process (certainly less than 1% of the total training time in all of our experiments). More formally,  the complexity of computing the weight for a certain example is $O( k * \log N_{\mathrm{val}} )$, where $k$ is the number of neighbors used, and $N_{\mathrm{val}}$ is the size of the validation dataset (typically quite small).
>
> - *If the labels from the teacher model are not reliable ... why not apply some label noise methods...*
>
> 	Our approach already incorporates techniques inspired by the learning with the noisy labels literature —  debiasing weighting schemes is a family of principled methods for dealing with label noise. Our contribution is to observe and exploit the fact that **the source of noise in distillation (the teacher model) is neither random nor arbitrary**, as it correlates with several metrics of confidence such as the margin-score, the entopy etc. We use and quantify this empirical observation to our advantage in order to formulate a principled method for constructing debiasing weights which comes with theoretical guarantees.
>
>
> - *In the experiments, the performance gain is not that much ...*
>
> 	Our method consistently outperforms the baselines. From a practical standpoint, it produces a principled,  fully "plug and play" distillation technique, that is composable with every other known distillation technique, and often provides significant improvements **essentially "for free"**.
>
> -  *The paper does not show how and why using K-nn to learn ...?*
>
> 	The reason we used kNN is because it is one the simplest and **computationally efficient** methods which does 	  not require any parametric assumption. Additionally, the number of neighbors can be determined from well-		established mathematical formulas, which makes the overall approach **hyperparameter-free** and easy to use.
>
> 	Other methods that could be used instead of kNN is essentially any other parametric regression approach (linear, polynomial etc), as long as one is confident about their choice of the appropriate parametric model.
>
>
> - *I'm wondering why or how likely the pseudo labels are not reliable ... this may be not a big issue.*
>
> 	Since the teacher model has not been trained on the unlabeled examples (by definition), it often generates inaccurate predictions for them. For example, if the teacher model has 90% accuracy, this means that the 10% of the pseudo-labels it generates are incorrect. Thus, even if the labeled and unlabeled data come from the same distribution, having, say, 10%-30% of the pseudo-labels generated from the teacher model being incorrect, can significantly harm the training process of the student. Indeed, this is a well-known problem in the literature of semi-supervised learning. In the revised version of our paper, we plan to include experiments which show that having an "oracle" that filters out the incorrect predictions of the teacher on the unlabeled examples can dramatically increase the performance of the student.
>
> - *It seems that the authors haven't addressed the potential negative societal impact of their work.*
>
> 	Knowledge-distillation  is a highly popular method for training Deep Neural Networks. As such, the potential negative societal impact of our work is related to the potential malicious usage of the Deep Learning technology. This is an important societal issue, as the latter has widespread applications that range from Natural Language Processing to Robotics and Self-Driving cars.  We plan to include the comments on the potential negative societal impact of our work in the revised version of our paper.

---

> > ### Author Response · Authors · 2022-08-09
> > **After Rebuttal**
> >
> > Dear Reviewer,
> >
> > since the end of discussion period is approaching, we would like to ask you whether our response helped in clarifying things and/or whether you have any other questions.
> >
> > Once again, thank you for your time and effort.
> >
> > Best Regards,
> >
> >  The authors

---

### Official Review · Reviewer_C5ER · 2022-07-10

**Rating:** 5
**Confidence:** 4
**Ethics Flag:** Yes
**Soundness:** 3 good
**Presentation:** 2 fair
**Contribution:** 3 good

**Summary:**

The authors propose a knowledge distillation technique for semi supervised settings. For the labeled data, they uniformly assign equal weights for the cross entropy loss between the predicted output and the supervision. On the other hand, the proposed method distributes adaptive weights for each data, which is theoretically and rigorously justified. The experimental results also support their theoretical analysis.


**Questions:**

1. It is hard to understand k-NN(U, Query) in the algorithm part 1. Could you explain it in detail? I can not find the definition of the  function.

**Ethics Review Area:**

["I don’t know"]

**Limitations:**

The authors adequately addressed the limitations and potential negative social impact of their work.

**Strengths And Weaknesses:**

Strengths:
1. The method is well-motivated and theoretically justified.
2. Even though the accuracy improvements are marginal compared to the baselines, the proposed method consistently outperforms them.

Weaknesses:
1. The method is hard to follow due to the missing details and confusing notations. For example, D has different meanings in algorithm 1 and line 183.
2. The experiments were insufficient. The previous standard knowledge distillation techniques such as [1,2,3] can still be applied for the problem settings used in the main paper. Could you compare with the previous algorithms [1,2,3]?

[1] Wasserstein Contrastive Representation Distillation, CVPR 2021.
[2] Contrastive Representation Distillation, ICLR 2020.
[3] Variational Information Distillation for Knowledge Transfer, CVPR 2019.

---

> ### Author Response · Authors · 2022-08-02
> **Response to reviewer C5ER**
>
> We thank the reviewer for their constructive feedback and suggestions.
>
> - *The method is hard to follow due to ... .*
>
> 	Regarding the presentation, we comit to implement all of the reviewers' comments in the next revision of our 	paper.
>
> - *The experiments were insufficient. ... compare with the previous algorithms [1,2,3]?*
>
> 	The relation of our work with the distillation methods proposed by the author are discussed in the next 	paragraph (**Composition with other distillation techniques**). This paragraph contains discussion and experimental results showing that our method is composable with (and therefore, independent of) the distillation techniques suggested by reviewer.
>
> - *It is hard to understand k-NN(U, Query) ...*
>
> 	Our Nearest Neighbor setting is a standard two-input two-output regression task. In particular, our Nearest Neighbor data structure is constructed as follows. Each example x of the validation set is assigned the two following pairs of points:
>
> 	(i) (teacher confidence at x, student confidence at x). This is the **covariate** of the regression task.
>
> 	(ii) (1, distortion at (x)) if the teacher correctly predicts the label of x, or (0, distortion at x), if 	the teacher does not correctly predict the label of x. This is the **response** of the regression task.
>
> 	The query corresponding to an unlabled example x' is of the form (teacher confidence at x', student 		confidence at x'). The kNN data structure  returns the average response over the k closest in euclidean distance pairs (teacher confidence at x, student confidence at x) in the validation set.
>
>
> We hope that the above clarify things, and we are lookig forward to address any follow-up questions by the reviewer. We plan to include the experimental results and the discussion above in the revised version of our paper.
>
>
>
>
>
> ### **Composition with other distillation techniques**
>
>
> At a high-level, the methods presented in [1, 2, 3] are based on techniques that enforce greater consistency between the teacher and the student. These methods enforce consistency not only between the teacher's predictions and student's predictions, but also between the representations learned by the two models. **However, for examples where the teacher is inaccurate, forcing the student to match the teacher is still harmful, and therefore   weighting the corresponding loss functions via our method is still applicable and beneficial.** We demonstrate this fact by implementing the method of Variational Information Distillation for Knowledge Transfer (VID) [3] and showing how it is indeed beneficial to compose it with our method. We chose the gender binary classification task of CelebA as benchmark, because it is known (see e.g. [6]) that the more advanced distillation techniques tend to be more effective when applied to problem with few classes.
>
> In the experiments below the teacher model is a MobileNet with depth multiplier 2, and the student model is a MobileNet with depth multiplier 1. The experimental set up is exactly the same as in our original submission (see section describing the CelebA experiments). We compare two methods:
>
> - Unweighted VID: We implement the loss described in equations (2), (4) and (6) of [3].
> -  Weighted VID: We perform importance reweighting to the standard VID loss function using our method.
>
>
>
> Labeled Examples | 10000 | 15000 | 20000 | 25000 | 30000 |
> |-------|--------|--------|-------|------| -------- |
>  |Teacher (**soft**)   | 91.59 | 93.76| 94.41 | 94.86 | 94.92 |
>  Weighted VID (Ours) | **94.35 +/- 0.11**| **95.01  +/- 0.17**| **95.73 +/-  0.04**| **95.89 +/- 0.08** | **96.11 +/- 0.08** |
>  Unweighted VID | 94.11 +/- 0.11 | 94.76 +/- 0.14 | 95.46 +/- 0.11 | 95.69 +/- 0.05 |  95.88 +/- 0.03|
>
>
> *Experimental results on CelebA: Weighted vs Unweighted Variational Information Distillation with **soft**-labels.*
>
>
> Labeled Examples | 10000 | 15000 | 20000 | 25000 | 30000 |
> |-------|--------|--------|-------|------| -------- |
>  |Teacher (**hard**)   | 91.59 | 93.76| 94.41 | 94.86 | 94.92 |
>  Weighted VID (Ours) | **94.36 +/- 0.13**| **95.07  +/- 0.21**| **95.71 +/-  0.03**| **95.83 +/- 0.02** | **96.13 +/- 0.08** |
>  Unweighted VID |  94.03  +/- 0.064 | 94.92 +/- 0.14 | 95.40 +/- 0.045 | 95.67 +/- 0.02  | 95.90+/- 0.03|
>
> *Experimental results on CelebA: Weighted vs Unweighted Variational Information Distillation  with **hard**-labels.*
>
>
> [6] Subclass distillation [Muller,  Kornblith, Hinton 2020]

---

> > ### Comment · Reviewer_C5ER · 2022-08-07
> > **After Rebuttal**
> >
> > The authors conveniently addressed my questions and some of my concerns are resolved.
> > However, I have a concern about the accuracy ranges sometimes are overlapped between weighted VID and unweighted VID.
> > Therefore I maintain my initial rating and I still lean to accept.

---

> > > ### Author Response · Authors · 2022-08-07
> > > **After Rebuttal Discussion**
> > >
> > > We  sincerely thank the reviewer for their suggestions and for carefully going through our response. We would like to point out  the following:
> > >
> > > (i) in the CelebA setting, the performance-gaps between Conventional (unweighted) Distillation and Unweighted VID are in the same ballpark with the performance-gaps of Weighted VID and Unweighted VID.  In fact, in some cases Unweighted VID does not even outperform the Conventional method.
> > >
> > > For reference, we have included in the table we shared earlier in our response the accuracies of the Conventional method (we used the results from Figure 4 of the paper) — see below.
> > >
> > > (ii) Our Weighted VID approach leads to **consistent** improvements over Unweighted VID that are beyond the corresponding standard deviations.
> > >
> > >
> > >
> > >
> > >
> > > Labeled Examples | 10000 | 15000 | 20000 | 25000 | 30000 |
> > > |-------|--------|--------|-------|------| -------- |
> > >  |Teacher (**soft**)   | 91.59 | 93.76| 94.41 | 94.86 | 94.92 |
> > >  Weighted VID (Ours) | **94.35 +/- 0.11**| **95.01  +/- 0.17**| **95.73 +/-  0.04**| **95.89 +/- 0.08** | **96.11 +/- 0.08** |
> > >  Unweighted VID | 94.11 +/- 0.11 | 94.76 +/- 0.14 | 95.46 +/- 0.11 | 95.69 +/- 0.05 |  95.88 +/- 0.03|
> > >  Conventional (unweighted) Distillation | 93.68 +/- 0.01 | 94.92 +/- 0.02% |  95.38 +/- 0.07   |  95.69 +/- 0.05  | 95.73 +/- 0.09    |

---

### Official Review · Reviewer_gJJP · 2022-07-10

**Rating:** 3
**Confidence:** 4
**Soundness:** 2 fair
**Presentation:** 3 good
**Contribution:** 2 fair

**Summary:**

This paper deals with the problem of few-shot knowledge distillation and proposes a method for weighing unlabeled examples based on the gaps between the true and distilling teachers. The obtained weights are employed for computing the loss function.

**Questions:**

I have no questions about this paper.

**Limitations:**

See the Strengths and Weaknesses part.

**Strengths And Weaknesses:**

[Strengths]

S1. The proposed method is simple and intuitive, which would draw the attention of people who are interested in applying it to their own tasks or applications.

S2. The current manuscript is basically well written and easy to follow.

[Weaknesses]

W1. The current experimental evaluation fails to demonstrate the effectiveness of the proposed method due to the following two reasons:

  (1) It lacks empirical comparisons with other related methods. In particular, [Dehghani+ ICLR2018] and [Kimura+ BMVC2018] incorporated example weighting for knowledge distillation.

  (2) According to the experimental setup presented in Section 3.1, both the teacher and student models have access to whole the training dataset, which is not a typical setting in the context of knowledge distillation. The teacher is usually trained with the full size of supervised training examples and the student is able to employ the same dataset as the teacher (fully supervised), a part of the supervised dataset plus additional unsupervised examples (few-shot) or only unsupervised examples (zero-shot). Due to this experimental setting, it is very difficult to understand whether the current baseline sufficiently works well or not.

W2. The proposed method requires additional supervised validation examples for computing the weights for unsupervised examples, which is a critical limitation of the proposed method. Existing methods for weighing examples do not rely on the availability of validation examples. If they are available for training, it seems better to employ them as additional training examples.

---

> ### Author Response · Authors · 2022-08-02
> **Response to reviewer  gJJP**
>
> Thank you for your feedback and helpful references. Please find our answers below.
>
> -  *It lacks empirical comparisons with other related methods. ....*
>
> 	The relation of our work to the uncertainy-based techniques proposed by the author are discussed in the next 		paragraph (**Composition and comparison with uncertainty-based techniques**).
>
> - *According to the experimental setup presented in Section 3.1, ...  which is not a typical setting in the context of knowledge distillation. ... it is very difficult to understand whether the current baseline sufficiently works well or not.*
>
> 	We respectfully disagree with the reviewer since  the experimental setup of section 3.1 is *exactly* the setup 	of distilaltion with unlabeled examples, see the original paper of Bucilua, Caruana and Mizil [9]. We refer the reviewer to the beginning of **Section 2.3: Our method** (*We consider the standard setting for distillation with unlabeled examples where ....*) for a detailed description of the setting we consider.
>
> - *... requires additional supervised validation examples for computing the weights ...*
>
> 	Our method does not require "additional" supervised validation examples in the following sense: we can choose to split the original pool of labeled examples to two sets, one to be used for pretraining the student, and one to play the role of "validation examples" used to estimate the weights. After the estimation has been completed, these examples can be merged back to the pool of labeled (and unlabeled) examples to be used for the training process of the student. In the submitted manuscript we treat the "validation dataset" as a completely independent hold out set (and do not  use it for training the student) to make our presentation as conceptually clear as possible. In the following paragraph, where we compare our results with the papers suggested by the reviewer, we do follow the above "split  and merge-back"-methodology so that every method we use for comparison has access to the exact same number of labeled examples.
>
> We hope that the above clarify things, and we are looking forward to address any follow-up questions by the reviewer. We plan to include the experimental results and the discussion above in the revised version of our paper.
>
> [9] Cristian Bucilua, Rich Caruana, and Alexandru Niculescu-Mizil. Model compression. [SIGKDD, 2006] ˇ

---

> > ### Author Response · Authors · 2022-08-02
> > **Composition and comparison with uncertainty-based techniques**
> >
> > ### **Composition and comparison with uncertainty-based techniques**
> >
> > Uncertainty based heuristics in the context of distillation with unlabeled examples are methods which try to downweight (or filter-out) teacher-labeled examples for which the teacher model is "uncertain" (a plethora of notions of defining and measuring "uncertainty" have been proposed in the literature, e.g. entropy, margin-score, dropout variance etc.) **We  emphasize that all these methods are completely independent of the student model (i.e., they *only depend on the teacher model*), and therefore can be viewed as *preprocessing steps* that can be combined with the student-training process we propose.** Indeed, the (teacher-)uncertainly-based methods cannot hope to unbias (i.e., "correct") the loss-function of the student model and, therefore, they are **provably** composable with our unbiasing technique. We experimentaly demonstrate this fact by implementing the uncertainty-based reweighting method presented in [4, 5] and showing how it is indeed beneficial to compose it with our method. For completeness, we also directly **compare** the weighting scheme of [4, 5] with our method (showing that it tends to outperform [4,5]).
> >
> >
> >
> >
> > Labeled Examples  | 8000  |10500 | 13000   | 15500 | 18000|
> > |-------|--------|--------|-------|------| -------- |
> > Teacher (soft)   | 67.55%  |   72.85%| 74.85%  |  77.63% | 78.40% |
> > Our method|  68.88 +/- 	0.15%  |  73.26 +/- 	0.18% | 75.00 +/- 	0.25% | 77.88 +/-	0.26% |  **78.65 +/-	0.28%**|
> > Fidelity weighting  | 69.31 +/- 	0.41% |   73.39 +/- 	0.44% | 74.69 +/-	0.31%  | 77.10 +/-	0.16% | 78.19 +/-	0.19% |
> >   Composition |  **70.06 +/- 0.16%** |  **74.19 +/- 0.2%** | **75.19 +/-	0.18%**  | **78.43 +/-	0.14%** |  78.55 +/- 0.84%
> >
> > *Comparison and composition of our method with fidelity-based importance reweighting: Experimental results on CIFAR-10 using soft-distillation.*
> >
> >
> >
> >
> >
> >
> >
> > Labeled Examples  | 8000 | 10500 | 13000 | 15500 | 18000
> > |-------|--------|--------|-------|------| -------- |
> > Teacher (soft)   | 44.70%  | 51.45%|  56.00%  | 58.70% | 61.82% |
> > Our method  | 46.18 +/- 0.25% |  53.15  +/- 0.16% |  **57.51 +/- 0.4%**  |  **59.65 +/- 0.4%**| 62.02 +/- 0.25% |
> >  Fidelity weighting  | 46.32 +/- 0.23%  | 52.92 +/- 0.20%  | 57.40 +/- 0.47%  | 59.48 +/-  0.177% |  61.39 +/- 0.09%|
> >   Composition |  **46.62  +/- 0.44%**|   **53.39 +/- 0.16%**|  57.11 +/- 0.41% |  **59.63 + /- 0.15%** | **62.30 +/- 0.26**%
> >
> > *Comparison and composition of our method with fidelity-based importance reweighting: Experimental results on CIFAR-100 using soft-distillation.*
> >
> >
> >
> >
> > #### Experimental set up.
> >
> > In the CIFAR-10 experiments below the teacher model is a MobileNet with depth multiplier 2, and the student model is a MobileNet with depth multiplier 1. In the CIFAR-100 experiments below the teacher model is a ResNet-110, and the student model is a ResNet56. Every method has access to the same number of labeled examples and **there exist no separate hold-out set**. (For our method we estimate the weights by  "splitting" and "merging back" the given pool of labeled examples as described earlier.)  We compare the following three methods:
> >
> > - Fidelity weighting scheme [4, 5]: For every example x we use the entropy of the teacher soft-label prediction
> > as an uncertainty/confidence measure, which we denote by $e(x)$. We then compute the exponential weights described in [4,5] as $w(x) = \exp(-e(x)/\bar{e})$, where $\bar{e}$ is the average entropy over all training examples. We also tried different measures of uncertainty such as the margin and variance of the teacher predictions which led to worse results.
> > - Our method using the same confidence metric (entropy) as in the fidelity weighting sceme. In the case of CIFAR-10 we recompute the weights every 50 epochs. In the case of CIFAR-100 the weights are computed only once in the beginning of the process.
> > - Composition of the two (multiplying the weights resulting from the two methods).

---

> > > ### Author Response · Authors · 2022-08-08
> > > **After Rebuttal**
> > >
> > > Dear Reviewer,
> > >
> > > since the end of discussion period is approaching, we would like to ask you whether our response helped in clarifying things and/or whether you have any questions.
> > >
> > > Once again, thank you for your time and effort.
> > >
> > > Best Regards,
> > > The authors

---

### Official Review · Reviewer_1d5d · 2022-07-12

**Rating:** 8
**Confidence:** 4
**Soundness:** 4 excellent
**Presentation:** 4 excellent
**Contribution:** 3 good

**Summary:**

Distillation is very popular technique both for preparing smaller student model for inference or doing some sort of pseudo-labeling including usage of unlabeled data. Current paper continues study in distillation topic and proposed a new theoretically motivated loss reweighting method to improve distillation of the teacher. The method is mainly studied in context of distillation on huge amount of unlabeled data. Here first student model is trained on supervised data, then the weights for every unlabeled sample are computed once from the results on validation data (we take KNN for distillation samples among validation data to tell what weight value will be), and finally we finetune model with weighted loss on mix of labeled and unlabeled data. Idea behind the weights is to have unbiased estimation for the risk minimization in distillation setting and there is an interesting discussion in the paper on noisy labels relation. Authors perform experiments on various datasets for image classification both with hard labels and soft labels with standard distillation or with a proposed method. Additional ablations on the supervised dataset size are conducted. Proposed method is hyper-parameter free, data-agnostic, and simple to implement. Experimental results are supported by theoretical analysis which rigorously justifies the method in certain settings.

**Questions:**

**Main paper**

The paper is very well written, thanks authors for this careful formulations and style of writing! Please find below my small suggestions on improving the text and typos as well as some extra questions on the results.
- I would suggest authors to include [1] into introduction / related works section and discuss this paper in the context of usage unlabeled data and getting unbiased predictions from teacher. Also would be a bit simpler to track references if they are ordered by their appearance in the text.
- line 27 "overparametrized networks" - here I wonder about this formulation (and do not agree with it) as student model is often much smaller (and authors also mentioned this before), so hard to say how this small student models will be overparametrized. Maybe reformulate this a bit?
- In the whole paper (including appendix) there is a bit mess with usage of space only for inputs or for pair (input, target), e.g. lines 129-130, equation (3), etc. Would be simpler to have a unified way on these definitions and usages in expectation formulas. Maybe having notation of $X$ and $\tilde{X}$ for input data saying that these spaces are the same for original data and adversarial.
- line 182 formula for $k$: fix $2 / 2+D$ to $2 / [2+D]$
- With respect to recent results in [1] I wonder what is happening with proposed method in case we have only labeled data and distillation is done as the original knowledge distillation paper (on training data of teacher model only)? Does reweighting scheme work? Also does unlabeled data improves distillation compared to distillation on supervised data only? [1] showed interesting results in this direction.
- [1] also introduced discrepancy between training quality (how we transfer knowledge from teacher to student) and generalization (about which we care). Would be interesting to see training curves (training loss and accuracy) for student models with standard distillation and weighted one as maybe proper weights change the optimization and simplify it even.
- Could authors provide what is approximate amount of time on computing weights for every sample? I guess it is negligible overall compared to the training, but curious to see confirmation.
- Do authors do any ablations on recomputing the weights for every sample say every 50 epochs? At least to know if it changes a lot or not during the training.
- Could authors perform ablation with temperature for the soft distillation case? Here we know (see e.g. in [1]) that temperature is giving a lot for optimization and generalization, even outweights other techniques to improve distillation. I wonder how it can change the optimization procedure and affect the proposed method. Also in practice we most likely use temperature not equal to 1.
- Could authors provide numbers on the teacher performance pre-trained on supervised data before starting the distillation?
- In Section 4 notation on spaces and distributions is not consistent with Section 2.

**Appendix**

I went through almost all appendix except last two proofs for sgd. General comment - Appendix needs a revision, it is less clear and careful prepared compared to the main text. For the rest of the text comments are below
- Appendix A plots are strange: teacher accuracies between hard and soft distillation are not consistent (e.g. Fig 8 leftmost top and bottom plots). Why are they different? We start distillation from the same teacher for both hard and soft distillation so teacher accuracy should be exactly the same.
- line 547 "if the sample" -> "is the sample"
- line 558-559 usage of $M$ while it should be $\mathcal{M}$
- Equations (7) and (8): what about variance estimation? Do we have any assumption on it? Or it is bounded for some reason?
- line 572: I repeated the math several times and got another range for $p$. It is $[0, (1-2d) / (1-d)^2]$ where $d$ is distortion. This does not affect the main conclusion and restricting weights to $[0, 1]$.
- C3 section should be moved earlier before C1
- in Equations after line 586 I don't understand how $l(f_{true}, f)$ disappears in $Bias(f)$. I think $Bias(f)=E_{x \sim\mathbb{P}} \\,p(x) (d - 1) * l(f_{true}, f)$ where $d$ is distortion.
- line 577 type "than" -> "that"
- notation is Section C3 and beginning of D should be fixed as again it is messed up with notation for input only and for (input, target) pairs.
- lines 600-606: usage of notation $z$ should be changed. It is used as $x$ context and later as $y$ context.
- line 629 - seems English language should be fixed here.
- It will be simpler to read if all proofs from Section D.* will be placed in the first statements formulation in section D itself, not reformulating this several times.
- Lemma D.7 - please add proper reference to the proof of this property (or via which theorem it can be proved).


[1] Stanton, S., Izmailov, P., Kirichenko, P., Alemi, A.A. and Wilson, A.G., 2021. Does knowledge distillation really work?. Advances in Neural Information Processing Systems, 34, pp.6906-6919.

**Limitations:**

Limitations are listed in the conclusion section.


**Discussion period updates**: I changed my rating score from 7 to 8 during the discussion period, details see in the thread below.

**Strengths And Weaknesses:**

**Strengths**
- A new reweighting method for knowledge distillation with unlabeled data which is based not on the uncertainty estimation (as people often do in pseudo-labeling) but on unbiased reweighting.
- The method is simple to implement
- Strong experimentation results across different datasets for image classification task (including ImageNet and smaller datasets): both for hard and soft distillation / with different amount of supervised data proposed method has statistically better results. [Sometimes however improvement is still marginal though statistically better].
- Interesting connection to noisy labels but here authors formulate the problem that noise is very specific and we can play with it.
- Detailed instructions on reproduction experiments which are simple to follow.
- Theoretical justifications of the proposed method for simpler forms of functions and standard regularity conditions: (i) it is information-theoretically optimal; and (ii) SGD optimization of the reweighted objective learns a solution with nearly optimal generalization.

**Weaknesses**
- Some mistakes in the proofs and formulations are found from Appendix section, however these findings should not affect the result and main statements.
- Additional results on distillation on supervised data and training curves for current results would be great to have for the full picture (details see in the Questions section).
- Ablation on recomputing the weights during the training (with some simple schedule at least, say every 50 epochs).
- Ablation on the temperature for soft distillation as it could be the main contribution for the generalization and maybe proposed approach doesn't change much. Temperature usage is very standard setting in distillation works.

---

> ### Author Response · Authors · 2022-08-02
> **Response to reviewer 1d5d**
>
>
> We would like to thank the reviewer for supporting our paper and for their insightful comments. Please see our responses below.
>
> -  *I would suggest authors ...  Also would be a bit simpler ....*
>
> 	We thank the reviewer for bringing up the paper by Stanton et. al. — we agree that it is very related and we plan to  include and discuss its results in the next revision of our paper.
>
> - We commit to implement all the reviewer's suggestions regarding improvements in the presentation of our paper. We particularly thank the reviewer for catching our typo in the proof of Proposition 2.1.
>
> - *With respect to recent results in [1] .... Does reweighting scheme work? ...*
>
> 	Our reweighting scheme is not designed for, and it would probably not provide substantial benefits in the case of the 	original distillation paper where the knowledge-transfer takes place on training examples of the teacher. The reason is that the accuracy of the teacher model on the training examples is almost 100% (as  DNNs are able to memorize the training set), which would result in all our weights to be almost equal to 1.
>
> - *Also does unlabeled data improves distillation compared to distillation on supervised data only?*
>
> 	Distillation with unlabeled examples can indeed significantly improve distillation on supervised examples only — see for example [7], which provided the SOTA for ImageNet when it appeared, and  where distillation with unlabeled examples was a crucial part of their pipeline. More generally, distillation with unlabeled examples is the most commonly used  training-paradigm in applications where one finetunes and distills from very large-scale foundational models such BERT, GPT3, MUM etc.
>
> - *[1] also introduced discrepancy between training quality ....*
>
> 	We most certainly agree with the reviewer. We plan to investigate connections to observations made in the paper by Stanton et. al. and report them in the revised version of our paper. One such connection is that agreeing with the teacher's prediction on the unlabled examples is not necessarily correlated (and potentially negatively correlated after some point) with the student's test-accuracy.
>
>
> -  *.. approximate amount of time on computing weights   ..*
>
> 	As we mentioned in our general response, the time needed for computing the weights is not very significant (less than 1% of the total training time).
>
> - *Do authors do any ablations on recomputing the weights for every sample say every 50 epochs? ....*
>
> 	We present ablations on recomputing the weight for every sample every 50 epochs in the next paragraph (**Further improvements via weights-recomputation during training**).
>
> - *Could authors perform ablation with temperature for the soft distillation case?.*
>
> 	We present ablations on temperature for CIFAR-100 in the next paragraph (**Ablation on the effect of temperature**).
>
>
> - *Also in practice we most likely use temperature not equal to 1.*
>
>  	Since hyperparameter optimization is rather difficult in application of distillation with unlabeled examples (as they are typically large scale), practitioners actually often use temperature of 1 as the default temperature for soft-labels — see e.g., [1, 2], and this is why we also made this choice.
>
> - *Could authors provide numbers on the teacher performance ....?*
>
>  	Our tables already provide the accuracy achieved by the teacher on the available supervised data before starting the distillation process. Does the reviewer perhaps mean here the accuracy the teacher would achieve if every example of our academic dataset was labeled? If so, these numbers are as follows, and we will include them in our paper:
>
> 	-  CelebA: 97.66%
>
> 	-  CIFAR10: 86.48%
>
> 	-  CIFAR100: 74.30%
>
> 	-  ImageNet:  73.50%
>
> 	 -  SVHN: 94.30%
>
>
>
> We hope that the above clarify things, and we are looking forward to address any follow-up questions by the reviewer. We plan to include the experimental results and the discussion above in the revised version of our paper.

---

> > ### Author Response · Authors · 2022-08-02
> > **Ablations**
> >
> > ### Further improvements via weights-recomputation during training.
> >
> > Here we investigate the benefits of recomputing the weights during training (every 50 epochs). In our experiments,  the teacher model is MobileNet with depth-mutliplier 2, while the student model is a MobileNet with depth-multiplier 1. The size of the validation set is 500 examples. The experimental set up is the same as in our original submission except for the fact that the random split of the academic datasets in labeled and unlabeled examples takes place once in the beginning of the experiment and is the same for all 3 trials (to reduce the variance of the experiment).
> >
> >
> >
> >
> > | Labeled Examples  | 8000  | 10500 |   13000 |  15500 |  18000|
> > |-------|--------|--------|-------|------| -------- |
> > Teacher (soft)   | 67.55%| 72.85% |  74.85 % | 77.63 % |  78.40  %|
> > Updating weights (per 50 epochs)|**69.19 +/- 0.32%** | **72.99 +/-	0.21%** | **75.08 +/- 0.13%** | **77.65 +/-	0.32%** |**78.42 +/-	0.15%**
> > One shot weights-learning | 68.76 +/-	0.14 % | 72.87 +/-	0.19 % |  74.39 +/-	0.10%  | 77.53	+/- 0.12 %| 77.89	+/- 0.13 % |
> >
> > *The effect of weights-recomputation on CIFAR-10.*
> >
> >
> > |Labeled Examples  | 8000  | 10500 | 13000  | 15500  |  18000 |
> > |-------|--------|--------|-------|------| -------- |
> >  | Teacher (soft)   |   87.55%  | 89.40% |  90.53% |  90.99% | 91.70% |
> > Updating weights (per 50 epochs)  | **90.82 +/-0.11%**  | **91.64 +/-0.36%**|   **92.82 +/-0.16%** |  **92.82 +/- 	0.44%**  |  **92.67+/-1.56%**|
> > One shot weights-learning | 88.02 +/- 	2.57%  | 89.42 +/-	2.07% |  90.72 +/- 1.96%  |92.19 +/- 	1.65% | **92.69 +/-	1.67%**  |
> >
> > *The effect of weights-recomputation on  SVHN.*
> >
> >
> >
> >
> >
> > ### Ablation on the effect of temperature.
> >
> > The experimental setting is identical to our original submission. The teacher model is a Resnet110 achieving test-accuracy $56 $%. The student model is a Renset56. The number of labeled examples is 12500.
> >
> > temperature|  unweighted|  weighted (ours)|
> > |--------|-------|---------------------------------|
> >    0.01   |      52.84 +/- 0.08   |      53.73 +/- 0.11|
> >    0.10    |     54.63  +/- 0.09  |      54.84 +/-0.12|
> >    0.50     |    56.45  +/- 0.12   |      57.01 +/- 0.1|
> >    0.80     |    56.67 +/- 0.12   |      57.60 +/- 0.15|
> >    1.00    |     57.17  +/- 0.15  |     57.56 +/- 0.09|
> >    2.00   |      57.54  +/- 0.11  |      **57.8 +/- 0.21**|
> >   3.00  |       57.20   +/- 0.18   |     57.09 +/- 0.25|
> >   5.00  |       56.92   +/- 0.11  |     57.01 +/- 0.2|
> >
> >
> > *The effect of temperature on CIFAR-100. *
> >
> >
> > [7] Self-training with Noisy Student improves ImageNet classification,   [XLHL, CVPR 2020]
> >
> > [8]  Meta-pseudo labels, [PDXL, CVPR 2021]

---

> > > ### Comment · Reviewer_1d5d · 2022-08-09
> > > **Respond to the comments (no further questions, all concerns are resolved)**
> > >
> > > Dear authors,
> > >
> > > Thanks a lot for going through all my comments and questions, additional comparisons and ablations!
> > >
> > > > Our tables already provide the accuracy achieved by the teacher on the available supervised data before starting the distillation process. Does the reviewer perhaps mean here the accuracy the teacher would achieve if every example of our academic dataset was labeled? If so, these numbers are as follows, and we will include them in our paper:
> > >
> > > Sorry for my bad formulation. I meant probably two things. One is what you provided here (the limit of teacher performance). Another is what student model can achieve if we train only on given labeled subset (so your pretraining of student). Probably for the full picture would be interesting to have student trained on everry academic dataset as labeled set too (similar what your reported here for teacher). This is just advise for the final revision for the full picture of the bounds on the performance.
> > >
> > > > Further improvements via weights-recomputation during training.
> > >
> > > Thanks for extra experiments! It is consistent with my expectations that recomputing will help. What is the most interesting here is how this helps for lower amount of supervision, where we can see the most significant improvement. This definitely makes sense, as with lower supervision the noise on pseudo-labels is higher from teacher + this noise more dominates during distillation, so dynamics of training is very different and adjusting more often the weights helps.
> > >
> > > > Ablation on the effect of temperature.
> > >
> > > Thanks for running different temperature ablations. Great to see that all improvements are very consistent, which experimentally demonstrates that proposed approach works robustly to temperature too and do proper job on weighting.
> > >
> > > > Appendix
> > >
> > > For the final revision please carefully rework appendix, and happy to have a look at it and recheck all proofs.
> > >
> > > **Update on my evaluation**
> > > I think with all provided extra experiments (regarding all reviewers concerns) and ablations (having consistent statistical improvements and composition property), theoretical proofs, simplicity and plug-in method property for wider practical adoption, the paper is very strong now, so I would like to raise my rating score from 7 to 8.

---

> > > > ### Author Response · Authors · 2022-08-09
> > > > **Thank you!**
> > > >
> > > > We greatly appreciate your feedback and effort in reading and evaluating our work. We will be happy to adopt all of your suggestions and rework our Appendix so that it summarizes our discussion.

---

### Author Response · Authors · 2022-08-02
**Thanking the reviewers**

We would like to thank all the reviewers for carefully reading our manuscript and providing insightful feedback.
We are encouraged by the positive feedback on: (i) the novelty of our work and its relevance to the NeurIPS community (1d5d, C5ER, HUb8); (ii) its simplicity and practicality (1d5d, gJJP); (iii) the theoretical justification of our method (1d5d, C5ER, HUb8); (iv) its consistency on outperforming the baselines (1d5d, C5ER); and (v) the writing quality, organization, and clean presentation of the ideas (1d5d, gJJP).

---

> ### Author Response · Authors · 2022-08-02
> **General Response**
>
> Here we address common questions raised by the reviewers.
>
>
> ## Comparison with other methods
>
> Reviewer C5ER  suggested comparing our method with the distillation techniques in [1, 2, 3], and reviewer gJJP suggested comparing our method with the uncertainty-based techniques in [4, 5].
>
> We point out that our method:
>
> - **(i)  is composable with (and, therefore, independent of) [1, 2, 3]**;
> - **(ii)  is composable with, but also typically outperforms [4, 5].**
>
>
>
> ###  **Composability with [1, 2, 3].**
>
> The methods [1, 2, 3] are based on techniques that enforce greater consistency between the teacher and the student (e.g. at the representation layers level). However, for examples where the teacher is inaccurate, forcing the student to match the teacher is  harmful, and therefore   weighting the corresponding loss functions via our method is  applicable and beneficial. We demonstrate this fact by implementing the method of Variational Information Distillation for Knowledge Transfer (VID) [3]. (See the discussion section dedicated to reviewer C5ER for more experiments and details.)
>
>
> Labeled Examples | 10000 | 15000 | 20000 | 25000 | 30000 |
> |-------|--------|--------|-------|------| -------- |
>  |Teacher (**soft**)   | 91.59 | 93.76| 94.41 | 94.86 | 94.92 |
>  Weighted VID (Ours) | **94.35 +/- 0.11**| **95.01  +/- 0.17**| **95.73 +/-  0.04**| **95.89 +/- 0.08** | **96.11 +/- 0.08** |
>  Unweighted VID | 94.11 +/- 0.11 | 94.76 +/- 0.14 | 95.46 +/- 0.11 | 95.69 +/- 0.05 |  95.88 +/- 0.03|
>
>
> *Experimental results on CelebA: Weighted vs Unweighted Variational Information Distillation with **soft**-labels.*
>
>
> ### **Composability and comparison with [4, 5].**
>
> Uncertainty based heuristics in the context of distillation with unlabeled examples are methods which try to downweight teacher-labeled examples for which the teacher model is "uncertain". These methods are  independent of the student model (they only depend on the teacher model), and can be viewed as preprocessing steps that can be combined with the student-training process we propose.  To demonstrate this:
> - We implement the uncertainty-based reweighting method presented in [4, 5] and show how it is indeed beneficial to combine it with our method.
> - We directly compare the weighting scheme of [4, 5] with our method (showing that ours tends to outperform).
>
> (See the discussion section dedicated to reviewer gJJP for more experiments and details.)
>
> Labeled Examples  | 8000 | 10500 | 13000 | 15500 | 18000
> |--|--|--|--|--| -- |
> Teacher (soft)   | 44.70%  | 51.45%|  56.00%  | 58.70% | 61.82% |
> Our method  | 46.18 +/- 0.25% |  53.15  +/- 0.16% |  **57.51 +/- 0.4%**  |  **59.65 +/- 0.4%**| 62.02 +/- 0.25% |
>  Fidelity weighting  | 46.32 +/- 0.23%  | 52.92 +/- 0.20%  | 57.40 +/- 0.47%  | 59.48 +/-  0.177% |  61.39 +/- 0.09%|
>   Composition |  **46.62  +/- 0.44%**|   **53.39 +/- 0.16%**|  57.11 +/- 0.41% |  **59.63 + /- 0.15%** | **62.30 +/- 0.26**%
>
> *Comparison and composition of our method with fidelity-based importance reweighting: Experimental results on CIFAR-100 using soft-distillation.*
>
>
> ## Computational efficiency / improving the accuracy.
>
> In this section we would like to clarify the computational aspects of our method in order to address questions raised by reviewers 1d5d and HUb8.
>
> -  The time for computing the weight for each example is insignificant compared to the actual training process (certainly less than 1% of the total training time in all of our experiments).
>
> -  The fact that the process of computing the weights is simple and inexpensive allows us to recompute them during training (say every 50 epochs, as suggested by reviewer 1d5d) to approximate the theoretically optimal  weights. See the discussion dedicated to reviewer 1d5d for mode details.
>
>
>
> | Labeled Examples  | 8000  | 10500 |   13000 |  15500 |  18000|
> |-------|--------|--------|-------|------| -------- |
> Teacher (soft)   | 67.55%| 72.85% |  74.85 % | 77.63 % |  78.40  %|
> Updating weights (per 50 epochs)|**69.19 +/- 0.32%** | **72.99 +/-	0.21%** | **75.08 +/- 0.13%** | **77.65 +/-	0.32%** |**78.42 +/-	0.15%**
> One shot weights-learning | 68.76 +/-	0.14 % | 72.87 +/-	0.19 % |  74.39 +/-	0.10%  | 77.53	+/- 0.12 %| 77.89	+/- 0.13 % |
> *The effect of weights-recomputation on CIFAR-10.*
>
>
>
>
>
>
>
> [1] Wasserstein Contrastive Representation Distillation [CWGLHC, CVPR 2021].
>
> [2] Contrastive Representation Distillation [TKI, ICLR 2020].
>
> [3] Variational Information Distillation for Knowledge Transfer [HDLD , CVPR 2019].
>
> [4] Fidelity-Weighted Learning  [DMGKS, ICLR2018].
>
> [5] Few-shot learning of neural networks from scratch by pseudo example optimization [KGTIU,  BMVC2018].

---

### Comment · Area_Chair_8q8Q · 2022-08-07
**Please engage author-reviewer discussion**

Dear Authors and Reviewers,

The deadline for author-reviewer discussion engagement is approaching. After that, authors and reviewers will be no longer able to discuss. Please post your questions if you have any, to clarify your paper from the authors and to express your concerns/questions from the reviewers.

For reviewers, if you have not already done so and there are author rebuttals to your review, please check the author rebuttal and response if it resolves your questions or not. Thank you.

Best
Your AC

---

### Meta-Review · Area_Chair_8q8Q · 2022-08-20

**Recommendation:** Accept
**Confidence:** Certain

**Metareview:**

This paper proposes to learn to reweight data samples in the distillation process to deal with potential noisy labels from the teacher. The writing is clear, and the empirical and theoretical results are satisfactory in general. At the beginning, the paper receives a mixture of positive and negative scores. The reviewers raise a number of questions, mostly about empirical comparisons with related works. The authors didi a good job in the rebuttal by providing many comparisons requested by the reviewers. The rebuttal resolves most of the concerns from the reviewers. Finally, the strong supporter remains strong support and the most negative one agrees that his concerns are also resolved, and urges that the authors should carefully revise the paper by incorporating all the results from the rebuttal into the paper. I standby the reviewers. Authors, please make sure to incorporate the extra results and clarifications into the final revision.

**Award:**

No

---

### Decision · Program_Chairs · 2022-09-14

Accept